# Dot1l interacts with Zc3h10 to activate Ucp1 and other thermogenic genes

**Danielle Yi[1,2†], Hai P Nguyen[1,2†], Jennie Dinh[1], Jose A Viscarra[1], Ying Xie[1], Frances Lin[1], Madeleine Zhu[1], Jon M Dempersmier[1], Yuhui Wang[1], Hei Sook Sul[1]***

[1]Department of Nutritional Sciences & Toxicology, University of California, Berkeley, Berkeley, United States; [2]Endocrinology Program, University of California, Berkeley, Berkeley, United States

**Abstract** Brown adipose tissue is a metabolically beneficial organ capable of dissipating chemical energy into heat, thereby increasing energy expenditure. Here, we identify Dot1l, the only known H3K79 methyltransferase, as an interacting partner of Zc3h10 that transcriptionally activates the *Ucp1* promoter and other BAT genes. Through a direct interaction, Dot1l is recruited by Zc3h10 to the promoter regions of thermogenic genes to function as a coactivator by methylating H3K79. We also show that Dot1l is induced during brown fat cell differentiation and by cold exposure and that Dot1l and its H3K79 methyltransferase activity is required for thermogenic gene program. Furthermore, we demonstrate that Dot1l ablation in mice using *Ucp1*-Cre prevents activation of *Ucp1* and other target genes to reduce thermogenic capacity and energy expenditure, promoting adiposity. Hence, Dot1l plays a critical role in the thermogenic program and may present as a future target for obesity therapeutics.

*For correspondence:
hsul@berkeley.edu

†These authors contributed equally to this work

**Competing interests:** The authors declare that no competing interests exist.

## Introduction

Adipose tissue has a central role in controlling mammalian energy metabolism in that, while white adipose tissue (WAT) is to store excess calories, brown adipose tissue (BAT) is to dissipate energy via non-shivering thermogenesis (*Cannon and Nedergaard, 2004*). Classic brown adipocytes contain a high density of mitochondria that contain Uncoupling protein 1 (Ucp1), which uncouples respiration from ATP synthesis and generates heat instead (*Farmer, 2008*; *Cannon and Nedergaard, 2004*). In addition, beige/brite adipocytes in WAT depot are recruited upon cold exposure or β3-adrenergic stimulation (*Seale et al., 2008*; *Wang and Seale, 2016*; *Sanchez-Gurmaches et al., 2016*; *Cypess et al., 2009*; *Lichtenbelt et al., 2009*; *Virtanen et al., 2009*; *Chondronikola et al., 2014*). Based on several cross-sectional studies, adult human brown fat or brown fat-like tissue is inversely correlated with body mass index and visceral fat (*Lichtenbelt et al., 2009*; *Hibi et al., 2016*; *Jimenez et al., 2007*). Therefore, unraveling the mechanisms underlying the thermogenic gene program has drawn growing attention in obesity research as a promising avenue to combat obesity and associated metabolic diseases.

One of the major advances in BAT biology has been understanding the transcription network that governs the thermogenic gene program and finding critical factors that activate the *Ucp1* gene. A multitude of transcriptional regulators have been implicated in the transcription of *Ucp1*, including transcription factors, Zfp516, Irf4, and Ebf2, and transcriptional coregulators, Prdm16, Ppargc1a, and Lsd1 (*Seale et al., 2008*; *Rajakumari et al., 2013*; *Dempersmier et al., 2015*; *Puigserver et al., 1998*; *Sambeat et al., 2016*; *Kong et al., 2014*). Recently, we identified a BAT-enriched and cold-induced transcription factor, Zc3h10 that activates *Ucp1* and other target genes, such as *Tfam* and *Nrf1* for mitochondrial biogenesis (*Yi et al., 2019*). Zc3h10 activates the *Ucp1* promoter by directly binding to the −4.6 kb region. Upon sympathetic stimulation, Zc3h10 is phosphorylated at S126 by p38 mitogen-activated protein kinase (MAPK) to increase binding to this distal region of the *Ucp1*

promoter. Consequently, Zc3h10 ablation in mice impairs the thermogenic gene program, while Zc3h10 overexpression in adipose tissue enhances the thermogenic capacity and energy expenditure, protecting mice from diet-induced obesity (*Yi et al., 2019*).

As in most biological processes, BAT as well as beige fat, rely heavily on environmental cues for its full activation of the thermogenic gene program. Integrating epigenetic effectors into the thermogenic network may provide a comprehensive understanding in the regulation of thermogenic gene program (*Yi et al., 2020*). Here, we identify Dot1l (disruptor of telemetric silencing 1-like) as an interacting partner of Zc3h10 and a critical coactivator of thermogenic genes. Dot1l is the only known methyltransferase that catalyzes the sequential mono-, di- and tri-methylation of H3K79, which, unlike major epigenetic sites, is located at the globular domain of nucleosome (*Min et al., 2003*; *van Leeuwen et al., 2002*; *Frederiks et al., 2008*). Differing from other histone methyltransferases, Dot1l does not contain a SET domain but uniquely has an AdoMET motif (*Feng et al., 2002*; *van Leeuwen et al., 2002*). Dot1l is broadly known to play roles in telomere silencing, cell cycle regulations and is particularly well studied in mixed lineage leukemia (MLL)-related leukemogenesis (*Schulze et al., 2009*; *Okada et al., 2005*; *Nguyen and Zhang, 2011*; *Ng et al., 2002*). However, not much is known about Dot1l recruitment to specific sites by specific transcription factors. Nor the role of Dot1l is thermogenic gene program is known.

We show here that Dot1l is recruited to the *Ucp1* promoter region via its direct interaction with Zc3h10 for Zc3h10-mediated transcriptional activation of *Ucp1* and other target genes. By using the specific chemical inhibitor of Dot1l-H3K79 methyltransferase activity, pinometostat (EPZ-5676), we demonstrate that Dot1l methyltransferase activity is required for thermogenic gene expression in vitro and in vivo. Moreover, Dot1l ablation in brown adipocytes impairs, while ectopic Dot1l expression enhances, thermogenic gene program and that Dot1l requires Zc3h10 for its function in thermogenesis. Dot1l ablation in *Ucp1*$^+$ cells in mice impairs the thermogenic capacity and lowers oxygen consumption, leading to weight gain. In this regard, the GWAS database reveals multiple SNPs of Dot1l associated with a waist-hip ratio and body mass index, further supporting a potential role of Dot1l in human obesity (GWAS Central identifier: HGVPM1111, HGVPM1114).

## Results

### Dot1l directly interacts with Zc3h10 for its recruitment and activation of *Ucp1* and other thermogenic genes

We previously reported Zc3h10 as a BAT-enriched transcription factor that promotes the BAT gene program (*Yi et al., 2019*). Most transcription factors do not work alone but rather form a complex to recruit other cofactors for transcription. Thus, as a DNA-binding protein, Zc3h10 may interact with coregulators to activate BAT gene transcription. Therefore, we searched for potential interacting partners of Zc3h10. The Zc3h10 with the streptavidin- and calmodulin-binding epitope-tag were incubated with nuclear extracts from BAT and the Zc3h10 complex after sequential purification were subjected to mass spectrometry. We identified multiple potential Zc3h10 interacting proteins, among which were methyltransferases and chromatin remodelers (*Figure 1—figure supplement 1A*). For further studying as a Zc3h10 interacting protein, we selected Dot1l (disruptor of telemetric silencing 1-like), the H3K79 methyltransferase, as it directly interacted with Zc3h10 and was the only BAT-enriched gene out of the candidates tested.

First, for validation of interaction between Dot1l and Zc3h10, we performed Co-IP experiments with lysates of HEK293 cells transfected with HA-*Dot1l* and Flag-Zc3h10 by using FLAG and HA antibodies. Indeed, we detected FLAG-Zc3h10 upon immunoprecipitation with HA antibodies. Conversely, HA-Dot1l was detected upon immunoprecipitation with FLAG antibodies (*Figure 1A*, left). We then tested an interaction of the two endogenous proteins using BAT from mice. Again, we detected the presence of endogenous Zc3h10 when nuclear extracts of BAT from mice were immunoprecipitated with Dot1l antibody. By reverse Co-IP, we confirmed the interaction between endogenous Dot1l and Zc3h10 (*Figure 1A*, right). We previously showed that Zc3h10 is recruited to thermogenic gene promoters by p38 MAPK in response to cold exposure/β$_3$ stimulation (*Yi et al., 2019*). To test if cold exposure/p38 MAPK can increase the interaction of Dot1l and Zc3h10, we took BAT lysates from mice housed at room temperature or 4°C and immunoprecipitated with either IgG or Dot1l and immunoblotted with Zc3h10. However, the interaction of Dot1l and Zc3h10 did

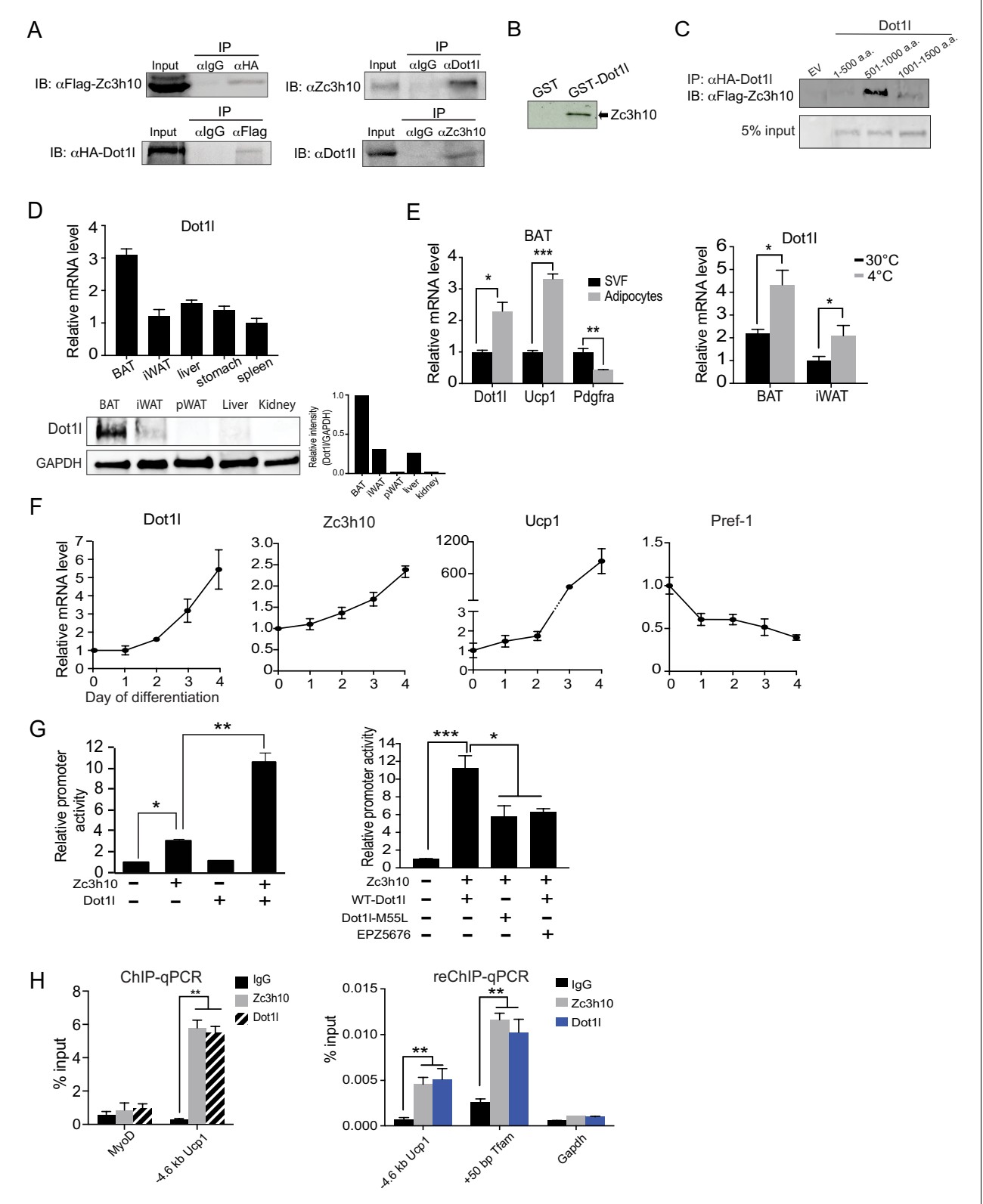

**Figure 1.** Dot1l directly interacts with Zc3h10 for its recruitment and activation of BAT gene program. (**A**) (Left) CoIP using αFlag for Zc3h10 or αHA for Dot1l after immunoprecipitation with either αHA- or αFlag, respectively, using lysates from HEK293FT cells transfected with Flag-*Zc3h10* and HA-*Dot1l*. (Right) CoIP of endogenous Zc3h10 and Dot1l protein using BAT tissue of C57BL/6 using αZc3h10 and αDot1l. (**B**) Autoradiograph of GST pull-down using GST-Dot1l and [35]S-labeled in vitro transcribed/translated Zc3h10. (**C**) CoIP using αFlag for Zc3h10 after immunoprecipitation with αHA for Dot1l

*Figure 1 continued on next page*

*Figure 1 continued*

using lysates from HEK293FT cells transfected with Flag-*Zc3h10* and various HA-*Dot1l* constructs. (**D**) RT-qPCR and western blot analysis of Dot1l in various tissues from 10-week-old C57BL/6 mice (n = 5). (Bottom right) Quantification of intensity of protein bands, calculated by *Dot1l/Gapdh*. (**E**) (Left) RT-qPCR for indicated genes in the adipocyte fraction and SVF from BAT. (Right) RT-qPCR for *Dot1l* mRNA of BAT and iWAT from mice housed at either 30˚C or 4˚C (n = 5). (**F**) RT-qPCR for indicated genes during the course of BAT cell differentiation. (**G**) (Left) HEK293 cells were cotransfected with the −5.5 kb *Ucp1*-Luc promoter with *Zc3h10* or *Dot1l* either together or individually (n = 5). (Right) Luciferase assay with HEK293 cells transfected with M55L-*Dot1l* or WT-*Dot1l* with the −5.5 kb *Ucp1*-Luc promoter with *Zc3h10*. (**H**) (Left) ChIP-qPCR for *Zc3h10* and *Dot1l* enrichment at the −4.6 kb region of *Ucp1* promoter using Flag or HA antibodies. Differentiated BAT cells were transduced with either Flag-*Zc3h10* or HA-*Dot1l*. (Right) ReChIP-qPCR for Dot1l and Zc3h10 co-occupancy using chromatin from BAT cells that were transduced with *Zc3h10* and/or *Dot1l*. Data are expressed as means ± standard errors of the means (SEM). *p<0.05, **p<0.01, ***p<0.001.

The online version of this article includes the following figure supplement(s) for figure 1:

**Figure supplement 1.** Dot1l directly interacts with Zc3h10 for UCP1 promoter activation.

not change between conditions, and cold exposure does not modulate the Zc3h10-Dot1l interaction (*Figure 1—figure supplement 1B*). We next asked whether Dot1l can directly bind Zc3h10, by using glutathione S-transferase (GST) fused to Dot1l, expressed and purified from *E. coli* (*Figure 1—figure supplement 1C*). Incubation of GST-Dot1l fusion protein immobilized on glutathione beads with in vitro transcribed and translated [$^{35}$S]-Zc3h10, but not control GST-alone, detected the presence of Zc3h10 (*Figure 1B*). Overall, these results demonstrate that Dot1l directly interacts with Zc3h10. We next examined domains of Dot1l for its interaction with Zc3h10. We generated three *Dot1l* truncated constructs; N-terminal Dot1l (1-500aa) containing the catalytic domain, the middle fragment (501-1000aa) containing the coiled-coil domain, and the C-terminal domain (1001-1540aa); All constructs were N-terminally tagged with HA. We then cotransfected these constructs with full-length Flag-tagged *Zc3h10*. By Co-IP, we detected the middle fragment (501–1000) of Dot1l, that contains coiled coil motifs, interacting with Zc3h10, but not the other two regions of Dot1l (*Figure 1C*). We conclude that the middle fragments of Dot1l contains the Zc3h10 interacting domain.

Since Zc3h10 activates *Ucp1* and other target genes for BAT gene program and Dot1l interacts with Zc3h10, Dot1l expression pattern could be similar to Zc3h10 and thus BAT-enriched. Indeed, tissue expression profiling by RT-qPCR and immunoblotting showed that Dot1l was enriched in mouse BAT compared to other tissues tested, such as WAT, liver, and kidney (*Figure 1D*). We next compared *Dot1l* mRNA levels between the stromal vascular fraction (SVF) that contains preadipocytes with adipocyte fraction of BAT from 10-wk-old wild-type (WT) mice. As expected, *Ucp1* was enriched in the adipocyte fraction, whereas *Pdgfra* expression was higher in the SVF fraction. We found *Dot1l* to be enriched in the brown adipocyte fraction by over 2-fold compared to the SVF (*Figure 1E*, left). Moreover, similar to *Zc3h10*, expression of *Dot1l* in BAT was induced upon cold exposure (*Figure 1E*, right). We next examined *Dot1l* expression during BAT cell differentiation in vitro. During the course of brown adipocyte differentiation, as expected, *Ucp1* expression was induced, whereas expression of preadipocyte gene, *Pref-1 (Dlk1)*, was suppressed. More importantly, similar to *Zc3h10*, *Dot1l* mRNA level was increased during BAT cell differentiation (*Figure 1F*). Overall, we conclude that Dot1l expression pattern is similar to Zc3h10 in terms of enrichment in mature brown adipocytes and induction by cold exposure, which may allow potential a cooperative function of Dot1l and Zc3h10 for the BAT gene program.

Next, to examine the functional significance of Dot1l and Zc3h10 interaction in the activation of the *Ucp1* promoter, we performed the luciferase (Luc) reporter assay using the −5.5 kb *Ucp1*-Luc promoter. Along with the *Ucp1* promoter-Luc construct, *Zc3h10* and *Dot1l* were cotransfected into HEK293 cells. As expected, compared to empty vector control, Zc3h10 alone activated the *Ucp1* promoter over 3-fold. Dot1l alone could not activate the *Ucp1* promoter. When co-transfected with Zc3h10, Dot1l was able to synergistically activate the *Ucp1* promoter, resulting in a robust 11-fold increase in the *Ucp1* promoter activity (*Figure 1G*, left). To test whether its enzymatic activity of Dot1l is required for activation of the *Ucp1* promoter, we generated a *Dot1l*-M55L construct, mutated of a critical residue in the catalytic core, located at its substrate, histone binding domain (*Zhu et al., 2018*). WT-*Dot1l* and *Dot1*-M55L were co-transfected with *Zc3h10* and the *Ucp1*-luc promoter into HEK293 cells. Cotransfection of WT-*Dot1l* along with *Zc3h10* increased the *Ucp1* promoter activity as expected. However, cotransfection of *Dot1l*-M55L along with *Zc3h10* and the *Ucp1*-luc significantly reduced the increase in the *Ucp1* promoter activity compared to WT-*Dot1l*

with *Zc3h10* (*Figure 1G*, right). Furthermore, we also treated HEK293 cells cotransfected with WT-*Dot1l* and *Zc3h10* with EPZ5676, a specific inhibitor of Dot1l H3K79 methyltransferase known to bind at the S-adenosyl methionine (SAM) binding pocket (*Daigle et al., 2013*). We detected significantly decrease in the *Ucp1* promoter activation by Dot1l and Zc3h10 (*Figure 1G*, right). These results demonstrate that catalytic activity of Dot1l is required for the cooperative function of Dot1l and Zc3h10 in the *Ucp1* promoter activation. Next, because Dot1l interacts with Zc3h10 to enhance the *Ucp1* promoter activity, we predicted that Dot1l should occupy the same *Ucp1* promoter region where Zc3h10 binds. We performed chromatin immunoprecipitation (ChIP) using BAT cells transduced with adenovirus containing Flag-tagged *Zc3h10* and HA-tagged *Dot1l*. By using Flag (for Zc3h10) and HA (for Dot1l) antibodies, we detected strong enrichment of both Zc3h10 and Dot1l at the - 4.6 kb region of the *Ucp1* promoter, a region corresponding to the Zc3h10-binding site (*Yi et al., 2019*; *Figure 1H*, left). We next performed a Re-ChIP experiment using Zc3h10 antibody and Dot1l antibody sequentially. Indeed, we observed significant enrichment of Dot1l for the −4.6 kb region of the *Ucp1* promoter as well as +50 bp *Tfam* promoter region, similar to Zc3h10 enrichment alone. However, there was no enrichment observed in the control, *Gapdh* promoter (*Figure 1H*, right). This illustrates the co-occupancy of Dot1l and Zc3h10 at the same regions of *Tfam* and *Ucp1* promoters. Collectively, we demonstrate that Dot1l, as an interacting partner of Zc3h10, is a critical coactivator of the *Ucp1* promoter.

## Dot1l is critical for the thermogenic gene program, and its action is dependent on Zc3h10

We next tested whether Dot1l is required for BAT gene program in cultured BAT cells. BAT cells at Day 2 of differentiation were transduced with adenovirus expressing short hairpin RNAs targeting *Dot1l* for knockdown (Dot1l KD) and were stimulated with forskolin for 6 hr on D6. Transduction of sh*Dot1l* adenovirus in BAT cells caused a decrease in *Dot1l* mRNA levels by approximately 60% (*Figure 2A*, left). While there were no apparent changes in expression of transcription factors critical for brown adipocyte differentiation, such as *Pparg*, there was a 50% reduction in *Ucp1* expression at mRNA and protein levels (*Figure 2A*). Expression of other Zc3h10 target genes, including *Tfam*, *Nrf1* and *Elovl3*, was also decreased significantly (*Figure 2A*, middle). Moreover, Dot1l KD BAT cells had decreased oxygen consumption rate (OCR) in oligomycin-, FCCP-, and rotenone/antimycin A-treated conditions (*Figure 2B*, left). Notably, the uncoupled respiration of Dot1l KD cells was significantly lower than the control BAT cells in both basal and in forskolin-stimulated conditions (*Figure 2B*, right). Additionally, we examined Dot1l's effect on adipogenesis to distinguish its effect on thermogenic gene expression and general regulation of adipogenesis. Indeed, there was no difference in expression of adipogenic marks, such as *Fabp4*, *Pparg*, *Adipoq* and no difference in lipid accumulation detected by Oil Red O staining (*Figure 2—figure supplement 1A*). These results show that Dot1l is critical for full activation of the BAT gene program and mitochondrogenesis, and thus thermogenic function in BAT cells.

Next, in order to further establish the role for Dot1l in BAT gene program, we treated BAT cells with a specific inhibitor of Dot1l H3K79 methyltransferase, EPZ5676 and stimulated the cells with forskolin for 6 hr on D6. We predicted that if Dot1l is required for the transcriptional activation of *Ucp1* and other BAT genes, inhibition of Dot1l activity should prevent BAT gene expression. Dot1l inhibition resulted in a significant reduction in mRNA and protein levels for Ucp1 by 90% (*Figure 2C*), while not affecting Dot1l protein stability (*Figure 2—figure supplement 1B*). Expression of other BAT-enriched Zc3h10 target genes, such as *Tfam*, *Nrf1*, *Cox7a*, and *Cox8b*, was decreased by 40–80%, whereas *Pparg* expression was not affected (*Figure 2C*). In order to confirm that the decrease in thermogenic gene expression is due to the enzymatic activity of Dot1l, we overexpressed Dot1l in differentiated BAT cells and treated with Dot1l inhibitor (10 uM), then performed ChIP-qPCR for H3K79me3. EPZ5676 treated cells had significantly decreased H3K79me3 at −4.6 kb *Ucp1* region as well as *Tfam* and *Nrf1* promoter regions, while having no difference at the *Gapdh* promoter region as compared to the control cells (*Figure 2D*). Furthermore, we overexpressed WT-Dot1l or M55L-Dot1l mutated of a critical residue in the catalytic core, in BAT cells and performed ChIP-qPCR. Dot1l-M55L mutant significantly reduced occupancy of H3K79me3 at the promoter regions of *Ucp1*, *Tfam* and *Nrf1*, compared to WT-Dot1l, while not affecting *Gapdh* promoter region (*Figure 2—figure supplement 1C*). These results show that Dot1l methyltransferase activity is required for thermogenic gene expression and other Zc3h10 target genes for the BAT gene program.

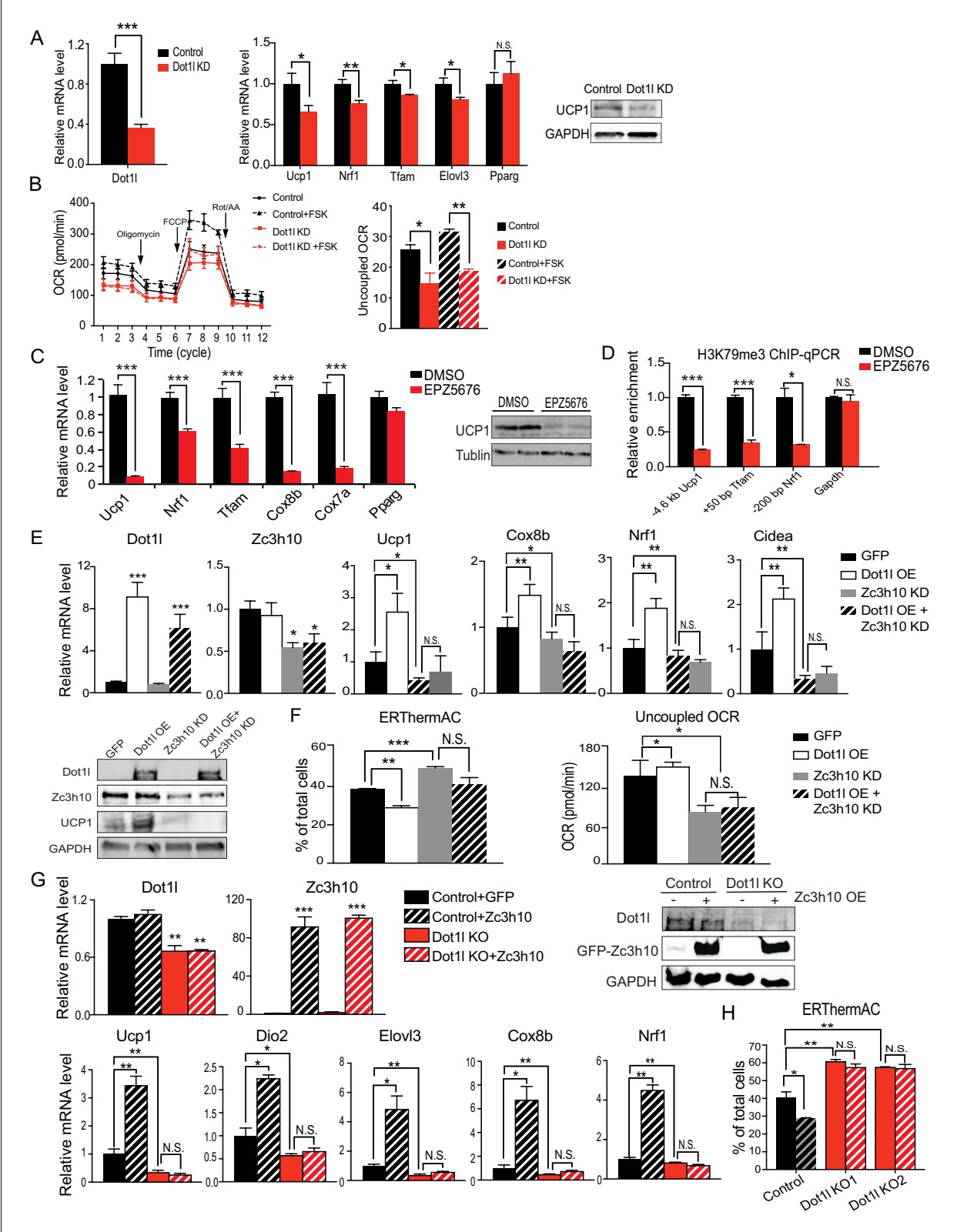

**Figure 2.** Dot1l is critical for the thermogenic gene program, and its action is dependent on Zc3h10. (**A**) (Left) RT-qPCR for *Dot1l* and thermogenic genes in BAT cells infected either scrambled (Control) or adenovirus expressing short hairpin targeting *Dot1l* (Dot1l KD) after D2 of adipogenic differentiation (n = 6). (Right) Immunoblotting for Ucp1. (**B**) (Left) OCR measured in Dot1l KD cells using Seahorse XF24 analyzer (n = 5). (Right) Uncoupled OCR in BAT cells infected with control or shDot1l under oligomycin (0.5 uM). (**C**) (Left) RT-qPCR for BAT cells treated with Dot1l chemical
*Figure 2 continued on next page*

*Figure 2 continued*

inhibitor, EPZ5676 (5 nM). (Right) Western blotting analysis for Ucp1 protein. (**D**) ChIP-qPCR for H3K79me3 at indicating genes using chromatin from differentiated BAT cells overexpressing Dot1l and treated with EPZ5676 (n = 5). (**E**) (Top) RT-qPCR for indicated genes and (Bottom) immunoblotting for indicated proteins in differentiated BAT cells that were transduced with either Ad*GFP* or Ad*Dot1l* or sh*Zc3h10* individually or in combination for overexpression of Dot1l and knockdown of Zc3h10 (n = 6). The differentiated cells were treated with forskolin (10 uM) for 6 hr. (**F**) (Left) FACS analysis and quantification of ERthermAC, that inversely correlates with heat production. (Right) Uncoupled OCR measured in BAT cells by Seahorse assay (n = 5). (**G**) RT-qPCR for indicated genes and immunoblotting for indicated proteins in the control BAT cells or *Dot1l*-CRISPR KO pool, overexpressing either GFP or Zc3h10. (**H**) FACS analysis and quantification of ERthermAC in Zc3h10 overexpressing in the control or *Dot1l*-CRISPR KO pools treated with forskolin(10 µM). Data are expressed as means ± standard errors of the means (SEM). *p<0.05, **p<0.01, ***p<0.001.

The online version of this article includes the following figure supplement(s) for figure 2:

**Figure supplement 1.** Dot1l promotes the thermogenic gene program in vitro.

To investigate whether Dot1l's function is dependent on its recruitment by Zc3h10 to thermogenic genes, we performed adenoviral overexpression of *Dot1l* (Dot1l OE) and knockdown of *Zc3h10* (Zc3h10 KD) alone or in combination in BAT cells on Day 2 of differentiation. We verified that *Dot1l* was overexpressed by over 6-fold, and *Zc3h10* was knocked down by 50% at the mRNA and protein levels (*Figure 2E*). As expected, Dot1l overexpression significantly increased thermogenic gene expression including *Ucp1*, *Cox8b*, *Cidea*, while Zc3h10 KD significantly decreased these genes (*Figure 2E*). To examine the functional changes of thermogenic capacity in these BAT cells, we utilized a small molecule thermosensitive fluorescent dye, ERthermAC in live cells (*Kriszt et al., 2017*). Corresponding to an increase in temperature, ERthermAC can accumulate in the ER to decrease fluorescence and, thus, fluorescence and thermogenesis are inversely correlated. Upon forskolin treatment, the population of ERthermAC$^+$ cells in Dot1l OE cells was decreased by 10%, which indicated that the cell population with higher temperature was increased by approximately 2.5-fold (*Figure 2F*, Left), calculated by the percent decrease from ERthermAC$^+$ population in control cells. In contrast, ERthermAC$^+$ cells in Zc3h10 KD cells were significantly increased by 10%, indicating that *Zc3h10* ablation decreased heat production in BAT cells. Moreover, Zc3h10 KD in Dot1l overexpressing cells showed low ERthermAC$^+$ cells that were similar to control cells, indicating that *Zc3h10* ablation prevented Dot1l-induced thermogenesis. Furthermore, when we measured OCR by using Seahorse, Dot1l overexpression in Zc3h10 KD BAT cells had the uncoupled OCR comparable to that of Zc3h10 KD, which was significantly lower than the control cells (*Figure 2F*, Right). These results are in accord with the concept that Dot1l activation of thermogenic genes is dependent on Zc3h10.

Next, we asked whether Dot1l is required for Zc3h10-induced thermogenesis. To induce Dot1l ablation in vitro, we used *Dot1l* knockout (Dot1l KO) pools generated by the CRISPR-inducible Cas9 system. We then overexpressed Zc3h10 in both control and the Dot1l KO pool and differentiated them. We verified that *Dot1l* was reduced at the mRNA levels and protein levels by approximately 50% in the Dot1l KO pool, and Zc3h10 was overexpressed by more than 80-fold. We found that Dot1l KO pool had significantly decreased *Ucp1*, *Dio2*, *Elov3*, *Cox8b*, and *Nrf1* mRNA levels by more than 50% compared to the control BAT cells, while Zc3h10 overexpression increased thermogenic gene expression as expected (*Figure 2G*). Importantly, Zc3h10 overexpression in Dot1l KO pool did not rescue the decreased BAT- gene expression, remaining in significantly reduced thermogenic gene expression, an evidence that Dot1l is critical for Zc3h10-induced thermogenic gene program. We also utilized ERthermAC to test heat generation in these cells. Zc3h10 OE significantly reduced ERthermAC$^+$ population in compared to control; while two independent Dot1l KO pools had increased ERthermAC$^+$ population, indicating decreased thermogenesis (*Figure 2H*). Notably, Zc3h10 OE did not affect ERthermAC$^+$ population in Dot1l KO pools. Altogether, these data further show that Dot1l is required for activation of thermogenic genes by Zc3h10.

We then tested the ability of Dot1l to induce *Ucp1* and other thermogenic genes by gain of function experiment using 3T3-L1 cells. We overexpressed Dot1l or Zc3h10 alone or together via adenoviral transduction in 3T3-L1 cells and the cells were induced to browning by forskolin treatment. We first verified Zc3h10 and Dot1l overexpression by RT-qPCR and immunoblotting (*Figure 2—figure supplement 1D*). Interestingly, Dot1l overexpression alone increased *Ucp1* mRNA levels over two-fold, probably due to the presence of endogenous Dot1l in 3T3-L1 cells. Co-overexpression of both Dot1l and Zc3h10 resulted in a further increase in *Ucp1* mRNA levels by three-fold and Ucp1 protein levels (*Figure 2—figure supplement 1D and E*). Furthermore, overexpression of Dot1l and Zc3h10

together in differentiated 3T3- L1 cells further increased mRNA levels for *Tfam* and *Nrf1* in comparison to Dot1l or Zc3h10 alone. Expression of a terminal adipose gene, FABP4, was not significantly different in all conditions. With increased expression of *Ucp1* and other BAT-enriched genes upon overexpression of Dot1l and Zc3h10 (*Figure 2—figure supplement 1D*), we next assessed the functional consequence of the oxygen consumption. Indeed, forskolin-treated Dot1l alone overexpressing cells had somewhat increased OCR. Co-overexpression of both *Dot1l* and *Zc3h10* further increased the total OCR as well as uncoupled OCR, suggesting cooperative function of Dot1l and Zc3h10 in browning of 3T3-L1 cells (*Figure 2—figure supplement 1F*). Altogether, these results demonstrate that Dot1l plays a critical role in promoting thermogenic program.

## Inhibition of Dot1l methyltransferase activity impairs BAT gene program in vivo

To study Dot1l function in vivo, we first used a selective and potent small molecule inhibitor of Dot1l, EPZ5676. We administered either saline (control) or EPZ5676 to WT mice via daily subcutaneous injection for 8 days (*Figure 3A*). Tissues were taken after 4 hr of cold exposure. By immunoblotting with H3K79me3 antibody, we confirmed inhibition of H3K79 methylation in BAT of animals treated with EPZ5676 (*Figure 3B*, left). Indeed, BAT from the inhibitor treated mice showed significantly decreased mRNA levels for *Ucp1*, *Nrf1*, and *Tfam*, as well as other thermogenic genes, such

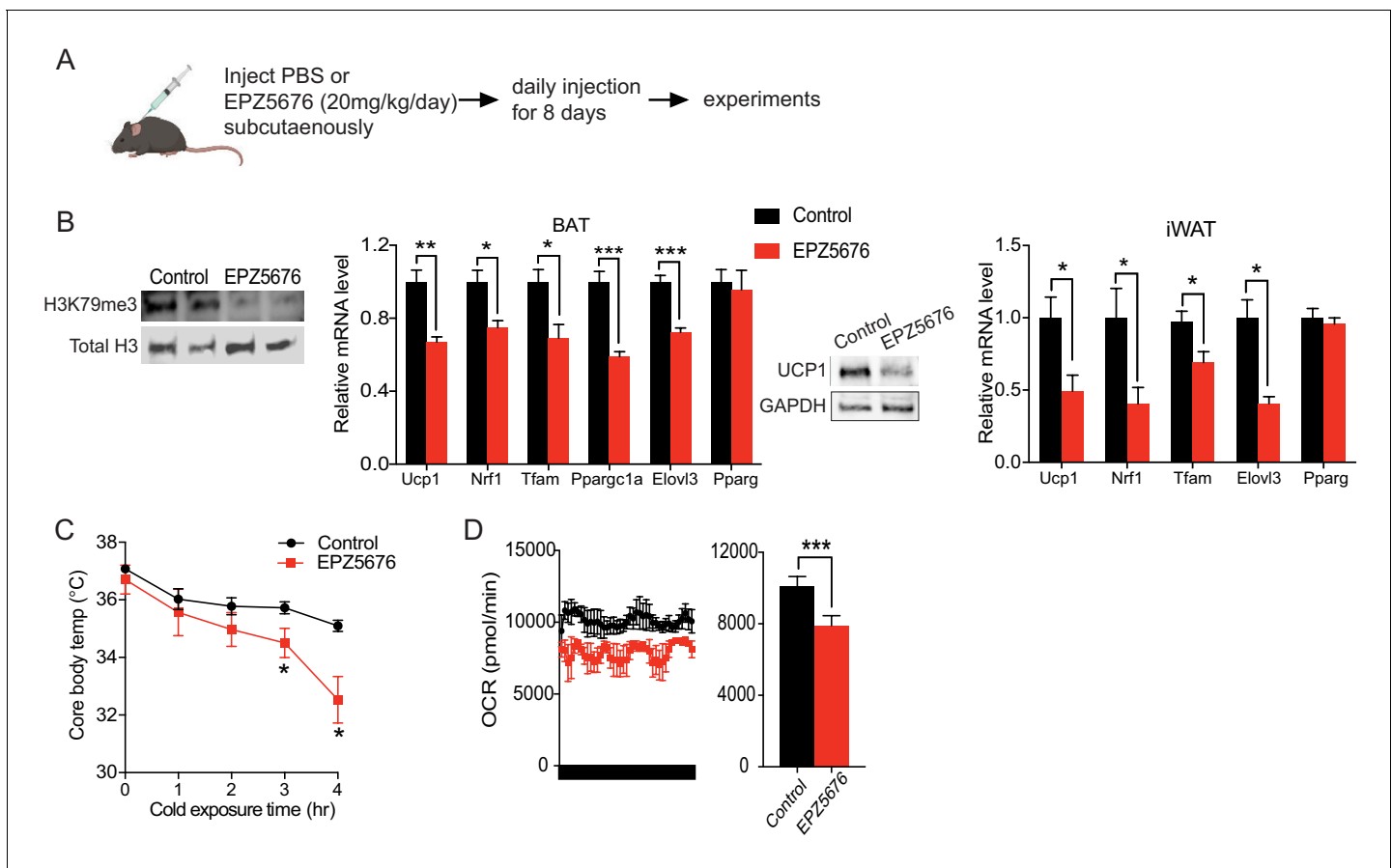

**Figure 3.** Inhibition of Dot1l activity impairs BAT gene program in vivo. (**A**) Schematic diagram of the strategy used to inject either Saline for control or Dot1l chemical inhibitor, EPZ5676. (**B**) (Left) Immunoblotting for H3K79me3 and total H3. (Middle) RT-qPCR for indicated genes in BAT and iWAT from either PBS or EPZ5676 injected mice (n = 4 per group) and immunoblotting for Ucp1 protein. (**C**) Rectal temperature of 14-week-old mice maintained at 4°C at indicated time points (hr) (n = 4 per group). (**D**) VO2 assayed in mice that were housed at 4°C by indirect calorimetry using CLAMS. Data are expressed as means ± standard errors of the means (SEM). *p<0.05, **p<0.01, ***p<0.001.

The online version of this article includes the following figure supplement(s) for figure 3:

**Figure supplement 1.** Inhibition of Dot1l activity impairs BAT gene program in vivo.

as *Ppargc1a* and *Elovl3*, while no changes in adipogenic transcription factor, such as *Pparg* (*Figure 3B*, middle). Also, the immunoblotting detected significantly lower Ucp1 protein levels in BAT of EPZ5676 treated mice compared to the control mice (*Figure 3B*, middle). Similarly, EPZ5676 treatment also decreased thermogenic gene expression in iWAT, WAT depot known to undergo browning (*Figure 3B*, right), but not in pWAT or in liver (*Figure 3—figure supplement 1A*).

Next, to investigate the physiological outcome from decreased expression of thermogenic genes, we subjected these EPZ5676 treated mice to acute cold exposure at 4°C. Both groups of mice had body temperatures around 37°C prior to the cold exposure. After 4 hr of cold exposure, the body temperature of the EPZ5676-treated mice was 4°C lower compared to their non-treated littermates, demonstrating the significantly reduced thermogenic capacity (*Figure 3C*). Next, we assessed the energy expenditure by measuring the whole body $O_2$ consumption using CLAMS. Indeed, EPZ5676 treated mice had significantly lower $VO_2$ than the control mice at 4°C (*Figure 3D*), whereas food intake and locomotive activity were similar between the two groups (*Figure 3—figure supplement 1B*). Altogether, these results support that Dot1l enzymatic activity is critical for activating the thermogenic gene program in vivo.

## The requirement of Dot1l for cold-induced thermogenesis and the Dot1l-Zc3h10 function in mice

To further evaluate whether Dot1l is critical for the BAT gene program in vivo, we performed *Dot1l* ablation in $Ucp1^+$ cells in mice (Dot1l-BKO) by crossing *Dot1l* floxed mice with *Ucp1*-Cre mice (*Figure 4A*, left). We validated our mouse model by genotyping and by comparing *Dot1l* mRNA levels in BAT and iWAT of Dot1l-BKO mice and *Dot1l* f/f control mice (WT). Dot1l-BKO mice had a 70% reduction in *Dot1l* mRNA and protein levels in BAT. Dot1l-BKO mice also showed a 60% decrease in *Dot1l* mRNA level in iWAT, probably due to the mild cold exposure at room temperature that can induce $Ucp1^+$ adipocytes in iWAT (*Figure 4A*, right). Importantly, *Ucp1* expression was significantly reduced by 40% and 60% in BAT and iWAT, respectively (*Figure 4B*). Ucp1 protein level also was reduced significantly, as detect in BAT (*Figure 4B* right). In addition, expression of other BAT enriched genes, such as *Dio2* and *Cidea*, and other Zc3h10 target genes, such as *Tfam* and *Nrf1*, were all significantly decreased in BAT and iWAT of Dot1l-BKO mice, compared to WT littermates (*Figure 4C*).

Dot1l is known to be the only known H3K79 methyltransferase (*van Leeuwen et al., 2002*). Hence, we performed ChIP-qPCR, to assess methylation status of H3K79 methylation at the Zc3h10 binding region of the *Ucp1* promoter using BAT of mice. Indeed, we detected a 50% reduction in di-methylation and a 70% reduction in tri-methylation of H3K79 at the −4.6 kb *Ucp1* promoter region (*Figure 4D*). Thus, the decreased *Ucp1* mRNA level upon Dot1l ablation in BAT was correlated with decreased di-methylation and tri-methylation H3K79 at the *Ucp1* promoter region. Similar to the −4.6 kb *Ucp1* promoter region, H3K79 methylation status at the −200 bp *Nrf1* promoter region and at the +50 bp *Tfam* region was also decreased (*Figure 4D*). We previous reported these regions to contain Zc3h10-binding sites. The reduced H3K79 methylation at Zc3h10 binding sites in BAT of Dot1l-BKO mice suggest that the recruitment of Dot1l by Zc3h10 to the target genes is important for BAT gene expression. Moreover, these observations support the notion that Dot1l participates in the transcriptional activation of *Ucp1* and other Zc3h0 target genes by modifying H3K79 methylation status.

To examine the effect of Dot1l deficiency on the thermogenic capacity, we subjected these Dot1l-BKO mice to an acute cold exposure at 4°C. While body temperatures of mice were similar initially, Dot1l-BKO mice were severely cold intolerant as their body temperature started to drop significantly after 2 hr of cold challenge. After 3 hr of cold exposure, the body temperature of Dot1l-BKO mice was 5°C lower than that of control mice (*Figure 4E*), an in vivo evidence of the requirement of Dot1l for cold-induced thermogenesis. Considering such lower thermogenic gene expression and decreased thermogenic capacity, we assessed the metabolic effect in the Dot1l-BKO mice by measuring whole body OCR using CLAMS. Indeed, the Dot1l-BKO mice had significantly reduced $VO_2$ compared to WT littermates at room temperature and at 4°C in particular, while locomotor activity and food consumption were similar (*Figure 4F* and *Figure 4—figure supplement 1A*). To examine the contribution of BAT to the altered energy expenditure in Dot1l-BKO mice, by Seahorse assay, we measured OCR in BAT dissected out from these mice. We found OCR in BAT from Dot1l-BKO mice to be decreased by 20%. (*Figure 4F*, right). In line with these results, we then asked whether

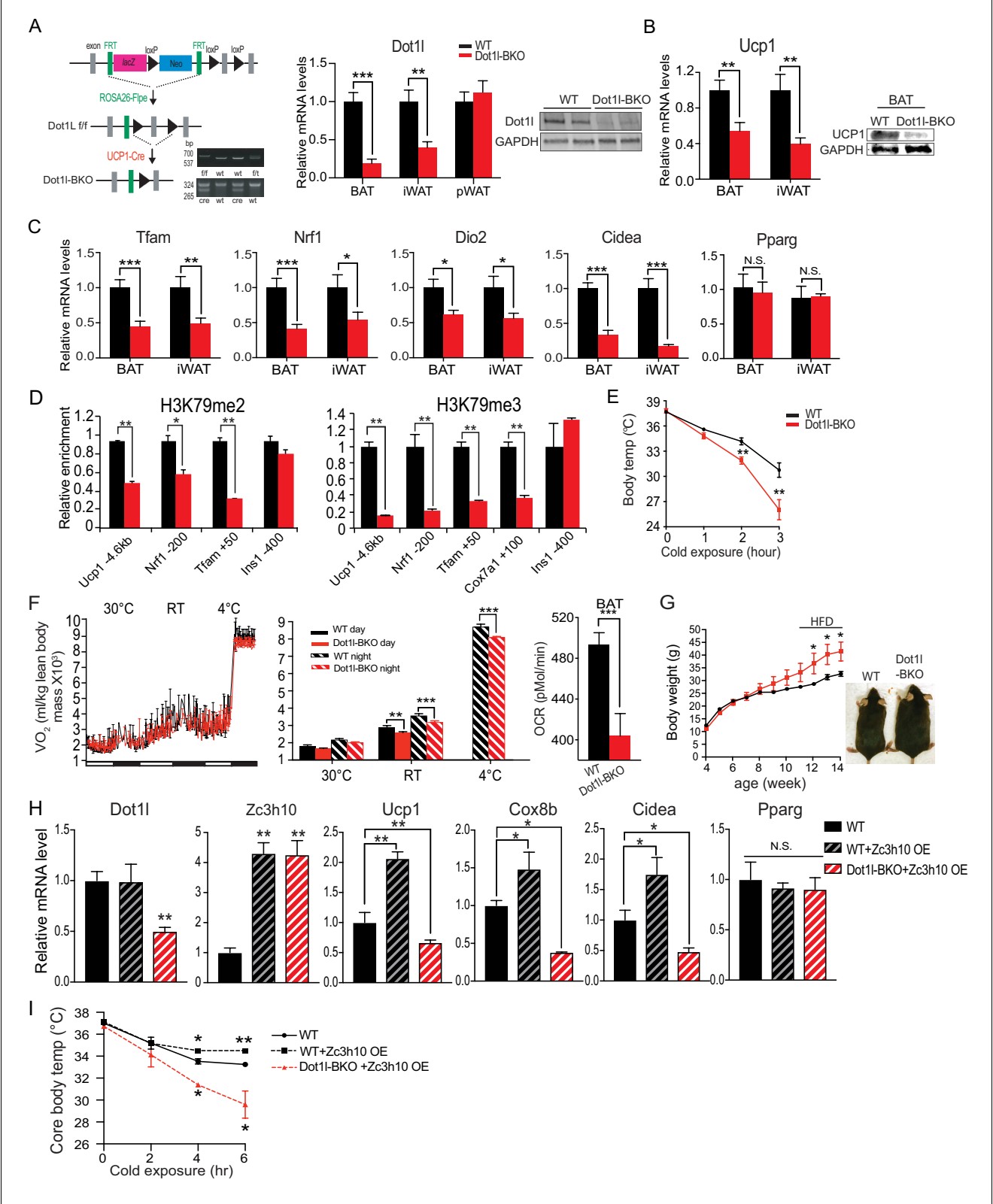

**Figure 4.** The requirement of Dot1l for cold-induced thermogenesis and the Dot1l-Zc3h10 function in mice. (**A**) (Left) Schematic diagram of the strategy used to generate BAT-specific *Dot1l* conditional knockout mice. PCR genotyping of the mice: Top gel, *Dot1l* allele; bottom gel, Cre. (Middle) RT-qPCR for *Dot1l* in BAT, iWAT and pWAT from Dot1l f/f (WT) and Dot1l-BKO mice (n = 6 per group). (Right) Immunoblotting for Dot1l in BAT from Dot1l f/f (WT) and Dot1l-BKO mice. (**B**) RT-qPCR and immunoblotting for Ucp1 in BAT and iWAT from Dot1l f/f (WT) and Dot1l-BKO mice. (**C**) RT-qPCR for

*Figure 4 continued on next page*

*Figure 4 continued*

indicated genes in BAT and iWAT from Dot1l f/f (WT) and Dot1l-BKO mice. (**D**) ChIP-qPCR of H3K79me2 and H3K79me3 at Zc3h10-binding regions of BAT tissue from control (f/f) or Dot1l-BKO mice (n = 5 per group). (**E**) Rectal temperature measured in 13-week-old mice at 4°C at indicated time points (hours) (n = 6 mice per group). (**F**) (Left) whole body VO$_2$ assayed in WT and Dot1l-BKO mice, housed at indicated ambient temperatures (n = 6 per group) by indirect calorimetry using CLAMS. (Right) OCR measured in BAT of WT and Dot1l-BKO mice using Seahorse XF24 Analyzer (n = 5 per group). (**G**) Representative photograph of 14-week-old control and body weight of WT and Dot1l-BKO mice fed HFD from wk 12. (**H**) RT-qPCR for indicated genes from WT injected with *GFP* or *Zc3h10* adenovirus or Dot1l-BKO mice injected with *Zc3h10* adenovirus for overexpression (n = 5 per group). (**I**) Rectal temperature measured at 4°C at indicated time points (hours) (n = 5 per group). Data are expressed as means ± standard errors of the means (SEM). *p<0.05, **p<0.01, ***p<0.001.

The online version of this article includes the following figure supplement(s) for figure 4:

**Figure supplement 1.** Dot1l is required for cold-induced thermogenesis in mice.

decreased energy expenditure and impaired BAT function in Dot1l-BKO mice would be reflected in changes in adiposity. The body weights of mice started to diverge starting on week 10, then they were maintained on a high-fat-diet (HFD) from week 11 to week 13 of age. By 13 weeks, the Dot1l-BKO mice were approximately 8 g heavier relative to the control mice (*Figure 4G*), and the Dot1l-BKO had significantly larger iWAT and pWAT depots, which primarily accounted for the higher total body weights of Dot1l-BKO mice (*Figure 4—figure supplement 1B*). There were no differences in food/energy intake between the two genotypes despite differences in body weights. In addition, blood glucose levels of Dot1l-BKO mice on HFD were significantly higher during the course of glucose tolerance test (GTT) (*Figure 4—figure supplement 1C*, Left), and they had significantly impaired insulin sensitivity when subjected to an insulin tolerance test (ITT) (*Figure 4—figure supplement 1C*, Right). Collectively, these results establish a defective BAT function and impaired thermogenic capacity that affects adiposity and consequently insulin sensitivity in Dot1l-BKO mice and that Dot1l is required for the BAT thermogenic program in vivo.

Next, to test the in vivo relevance of Dot1l-Zc3h10 interaction to thermogenesis, we overexpressed Zc3h10 in WT and in Dot1l-BKO mice by direct injection of *Zc3h10* adenovirus into BAT. We detected a four-fold increase in *Zc3h10* mRNA levels in BAT of *Zc3h10* adenovirus-injected mice (Zc3h10 OE). As expected, Dot1l-BKO mice showed a 60% reduction in Dot1l mRNA levels in BAT (*Figure 4H*). As we previously have reported that overexpression of Zc3h10 in adipose tissue enhances thermogenic gene expression (*Yi et al., 2019*), Zc3h10 overexpressing mice by injection had increased expression of BAT-enriched genes, such as *Ucp1*, *Cox8b*, and *Cidea*, compared to the control mice. More importantly, expression BAT-enriched genes remained significantly reduced in Dot1l-BKO even upon Zc3h10 overexpression, while adipogenic markers, such as *Pparg*, remained unchanged (*Figure 4H*). Moreover, when these mice were subjected to cold exposure, Zc3h10 OE mice had higher core body temperature compared to the control mice after 4 hr (*Figure 5I*). However, Dot1l-BKO Zc3h10 OE mice were severely cold sensitive as their body temperature was significantly lower than the control and Zc3h10 OE mice. Taken together, these results demonstrate the requirement of Dot1l-Zc3h10 interaction, both in vitro and in vivo for activation of the thermogenic program in brown adipose tissue.

## Genome wide analysis for Dot1l effect on chromatin accessibility for thermogenic gene program

Thus far, we have shown that Dot1l-BKO mice have severely reduced thermogenic capacity due to significantly decreased BAT gene expression accompanied with reduced H3K79me2/3 at promoter regions of thermogenic genes. To study the underlying mechanism of how Dot1l ablation in BAT decreases H3K79me2/3 to decrease chromatin accessibility, thereby decreasing gene transcription, we used nuclei isolated from cold exposed Dot1l-BKO mice and their littermates to assess the chromatin landscape during thermogenesis by Assay for Transposase-Accessible Chromatin (ATAC-seq). Strikingly, our analyses found approximately 20,000 peaks per condition (WT and Dot1l-BKO). We detected a high number of peaks in the aggregate plot (upper) and high ATAC-seq read distributions concentrated at the TSS in the heatmap for the WT (lower), while Dot1l-BKO had significantly reduced peaks near the transcription start sites (TSS), signifying reduced open chromatin regions, as H3K79 methylation has been reported as an activation marker. (*Figure 5A*, left). To gain comprehensive understanding of the genome wide chromatin structure changes, we examined the pathways

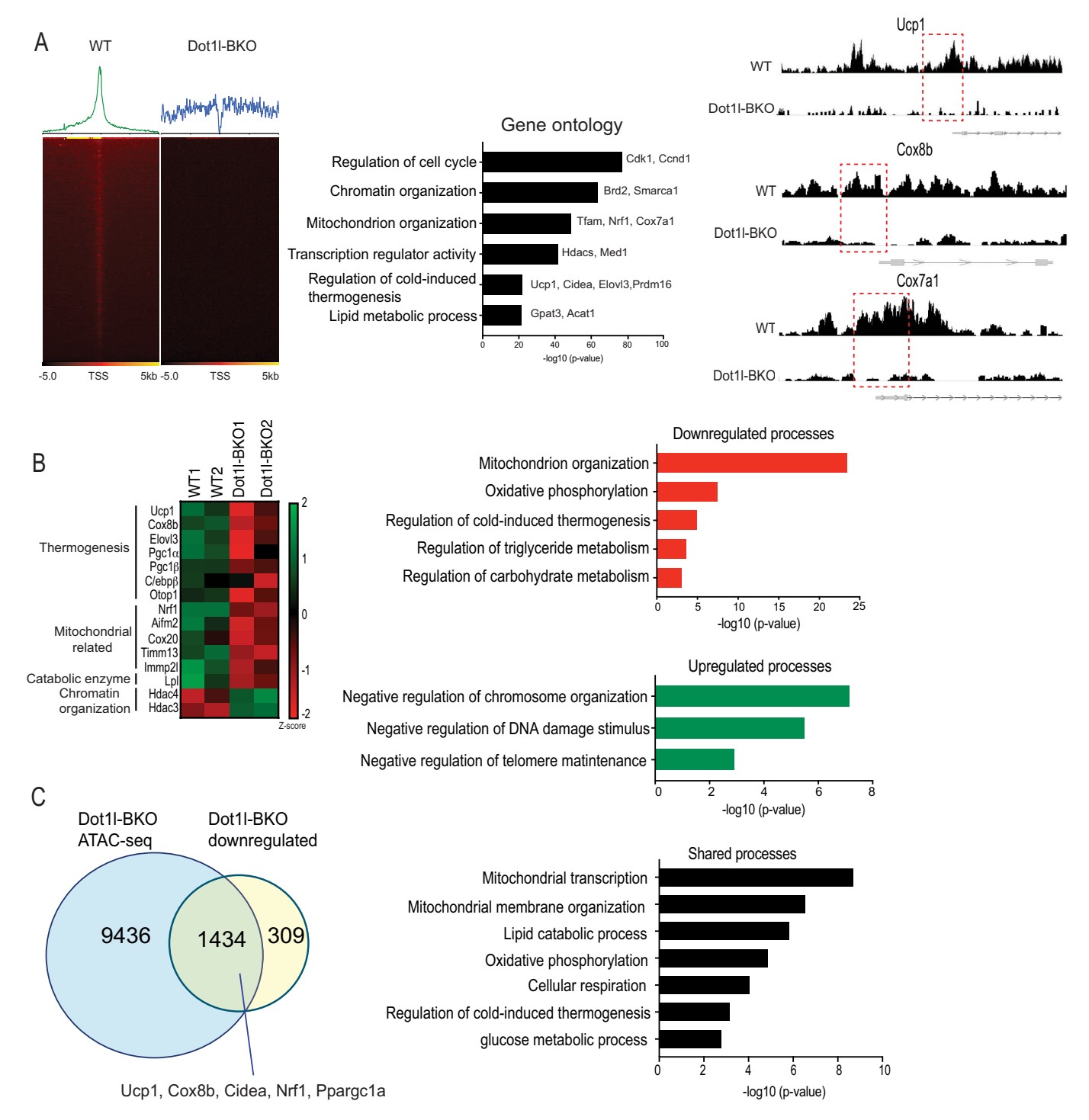

**Figure 5.** Genome wide analysis for Dot1l effect on chromatin accessibility for thermogenic gene program. (**A**) ATAC-seq using BAT from WT and Dot1l-BKO mice after 4 hr cold exposure n = 2 per group. (Left) Heatmaps showing open chromatin regions focused at the transcription start site (TSS). (Middle) Gene Ontology analysis of pathways that are affected by *Dot1l* ablation in BAT. (Right) UCSC genome browser screenshot of representative peaks at a subset of promoter regions of thermogenic genes. (**B**) RNA-seq using BAT from WT and Dot1l-BKO mice, n = 2 pooled RNA samples per group. (Left) Heatmap showing changes in gene expression. Color scale shows changes in gene expression as determined by Z-score, green is –two and red is 2. (Middle) Representative top GO terms of upregulated and downregulated genes identified by differential expression analysis. (**C**) Venn diagrams showing number of unique or shared genes between ATAC- and RNA-seq datasets and charts for representative top gene ontology (GO) terms.

that are affected by Dot1l ablation in BAT using Gene Ontology. The most affected pathways were cell cycle, chromatin organization, mitochondrion organization, thermogenesis etc (*Figure 5A*, middle). With our primary interest in assessing Dot1l's role in the regulation of thermogenesis, we compared relative ATAC-seq peaks for thermogenic genes, such as *Ucp1*, *Cox8b*, and *Cox7a*, in response to Dot1l ablation (*Figure 5*, middle). Dot1l-BKO mice had much smaller peaks around promoters of *Ucp1*, *Cox8b*, and *Cox7a* compared to WT mice (*Figure 5A*, right), consistent with significantly decreased thermogenic gene expression in the Dot1l-BKO mice (*Figure 4C*). These data suggest that H3K79me2/3 by Dot1l is important for creating a chromatin landscape permissive to transcription in BAT upon cold stimulation.

Next, to examine gene expression changes upon Dot1l ablation in *Ucp1*$^+$ cells as a result of chromatin compaction, we performed RNA-seq using BAT from Dot1l-BKO mice and their littermates. Gene ontology of genes that were at least two-fold downregulated in Dot1l-BKO mice revealed that mitochondrion organization, oxidative phosphorylation and cold-induced thermogenesis were some of the most affected processes (*Figure 5B*, right). Notably, the downregulated thermogenic genes include *Ucp1*, *Cox8b*, *Elovl3*, and *Ppargc1a* as well as those related to mitochondrion organization and oxidative phosphorylation such as *Nrf1*, *Cox20*, *Tomm13*, and *Immp21* (*Figure 5B*, left). Also, we found Dot1l-BKO had decreased expression of *Aifm2*, a recently reported BAT-specific enzyme that converts NADH to NAD to sustain robust glycolysis to fuel thermogenesis (*Nguyen et al., 2020*). In addition, regulation of triglyceride and carbohydrate catabolism were compromised by Dot1l ablation. Conversely, chromosome organization, negative regulation of telomere maintenance, response to DNA damage were upregulated in the Dot1l knockout. In fact, we detected upregulated genes involved in chromatin compaction and gene repression, such as *Hdac3*, *Hdac4*, and *Dnmt1*, further supporting that Dot1l increases chromatin accessibility (*Figure 5B*, right). With ATAC-seq results, these data suggest a mechanistic link between Dot1l-mediated chromatin remodeling and the global landscape of genome accessibility in brown fat for thermogenic gene expression.

We then compared RNA-seq data to ATAC-seq obtained to identify common downregulated pathways found at lower levels using ATACs upon Dot1l ablation in *Ucp1*$^+$ cells, and we found about 82% of the total of downregulated genes were also detected from the ATAC-seq dataset (*Figure 5C*, left). Some of similar processes between the two datasets include regulation of cold-induced thermogenesis, mitochondrial transcription and membrane organization, oxidative phosphorylation, and lipid catabolic process, (*Figure 5C*, right). Remarkably, the cluster of shared genes included thermogenic genes, such as *Ucp1*, *Ppargc1a*, *Cox8b*, *Nrf1*, and *Cidea*, further underscoring the importance of Dot1l in brown adipose tissue. This demonstrate that Dot1l ablation results in reduced chromatin accessibility, leading to downregulated gene expression for those specific pathways. Altogether, we conclude that H3K79me2/3 by Dot1l plays a critical role in thermogenic gene activation by increasing chromatin accessibility.

## Discussion

Numerous studies have supported the concept that a network of transcription factors and epigenetic coregulators must work together to fine-tune thermogenic program in response to environmental conditions (*Sambeat et al., 2017*). Here, we identify Dot1l, the only known H3K79 methyltransferase, as an interacting partner of Zc3h10, a transcriptional factor that activates *Ucp1*, as well as *Tfam* and *Nrf1*, for thermogenesis. We show that Dot1l is recruited by Zc3h10 to the *Ucp1* and other thermogenic genes. We demonstrate that Dot1l directly interacts with Zc3h10 to co-occupy the same region of the *Ucp1* promoter. Zc3h10 binds to Dot1l via the 501-1000aa fragment, which is largely composed of coiled coil domains, evolutionarily conserved and widely known for protein-protein interactions (*Mier et al., 2017*). This same coiled coil domain has been reported to interact with some of mixed lineage leukemia (MLL) oncogenic fusion proteins, such as AF10, to induce H3K79 methylation and constitutively activate a leukemic transcription program (*Song et al., 2019*). Here, we show that Dot1l is required for Zc3h10's function for thermogenic gene program. In fact, Zc3h10 overexpression could not rescue the decreased thermogenic gene expression upon Dot1l ablation in mice, demonstrating that Dot1l is critical in Zc3h10-mediated activation of thermogenesis.

We establish that H3K79me2/3, catalyzed by Dot1l, as an activation mark for thermogenic gene program. In this regard, previous studies reported that H3K79 methylation is strongly associated

with active transcription (*Steger et al., 2008*; *Wood et al., 2018*). Our Dot1l-BKO mice showed a markedly less open chromatin regions in promoter regions of thermogenic genes. The ATAC-seq data showed significantly reduced peaks genome-wide near the transcription start sites (TSS) in BAT of Dot1l-BKO mice, compared to control mice that had concentrated open chromatin regions at TSS upon cold exposure. In addition, studies reported that H3K79me2/3 may be active enhancer marks and may even be required for enhancer-promoter interactions to increase transcription (*Markenscoff-Papadimitriou et al., 2014*; *Gilan et al., 2016*; *Godfrey et al., 2019*). *Godfrey et al., 2019* found H3K79me2/3 to be abundant at a subset of super-enhancers in MLL-related leukemia cells and that a loss of H3K79me2/3 by DOT1L inhibition led to significantly reduced enhancer-promoter interaction coupled with a decrease in transcription (*Godfrey et al., 2019*). We show that Dot1l is recruited by Zc3h10 to the −4.6 kb upstream region of *Ucp1*, a region suggested to be one of so-called super enhancers (SEs) (*Whyte et al., 2013*; *Harms et al., 2015*). In fact, Dot1l ablation in BAT of mice resulted in a significantly reduction of H3K79me2/3 at the −4.6 kb *Ucp1* region, correlating with decreased expression of *Ucp1*. It is possible that H3K79me2/3 catalyzed by Dot1l may be important in maintaining enhancer/promoter association for transcriptional activation.

Due to limited accessibility of the H3K79 residue, located in the nucleosome core, how H3K79 methylation promotes gene activation is not clearly understood (*van Leeuwen et al., 2002*; *Lu et al., 2008*). Several studies support the concept that H3K79me2/3 and other well-profiled gene activation marks are interdependent (*Steger et al., 2008*). Steger et al. demonstrated that H3K79me2/3 at gene promoters highly correlates with H3K4me1/2/3 and H3K36me3, known gene activation marks in mammalian cells. Furthermore, Chen et al. reported that DOT1L may inhibit the localization of SIRT1 and SUV39H1, a H3K9ac demethylase and a H3K9me2 methyltransferase, respectively, at MLL fusion target genes correlating with elevated H3K9ac and low H3K9me2, to maintain an open chromatin state (*Chen et al., 2015*). These authors proposed that H3K79me2/3 mark, in part, may function by inhibiting the histone deacetylase activity (*Chen et al., 2015*; *Kang et al., 2018*). In line with these findings, Dot1l probably works with other histone modifiers to maintain activation marks or to suppress repressive marks. Regardless, we establish that H3K79me2 and H3K79me3, in particular, function as activation marks for thermogenic gene program.

Overall, we demonstrate that Dot1l is critical for thermogenic program through H3K79 methylation and the requirement of Dot1l-Zc3h10 interaction for brown adipose thermogenesis in vitro and in vivo. With combined ATAC-seq and RNA-seq data, we provide molecular evidence that the decreased thermogenic gene expression is due to decreased H3K79me2/3 on promoter regions of thermogenic genes, as well as genes involved in chromatin remodeling. Consequently, our mouse line of *Dot1l* ablated in Ucp1[+] cells show significantly decreased expression of thermogenic genes, including *Ucp1*, resulting in cold intolerance with decreased oxygen consumption (*Figure 4C–G*), leading to increased adiposity as well as insulin insensitivity.

# Materials and methods

### Key resources table

| Reagent type (species) or resource | Designation | Source or reference | Identifiers | Additional information |
| --- | --- | --- | --- | --- |
| Gene (*Mus musculus*) | *Dot1l* | GenBank | ID: 208266 | |
| Strain, strain background (*Escherichia coli*) | BL21(DE3) | NEB | C2527H | |
| Strain, strain background (*Mus musculus*) | ES cell line Dot1ltm1a(KOMP) | KOMP Repository | MMRRC:054749-UCD | |
| Genetic reagent (*Mus musculus*) | *Dot1l*-G55L | This lab | | |
| Cell line (*Homo sapien*) | HEK293 | UCB Cell Culture Facility | RRID:CVCL_0045 | Authenticated using STR analysis |

*Continued on next page*

*Continued*

| Reagent type (species) or resource | Designation | Source or reference | Identifiers | Additional information |
|---|---|---|---|---|
| Cell line (*Mus musculus*) | 3T3-L1 | UCB Cell Culture Facility | RRID:CVCL_0123 | Authenticated using STR analysis |
| Cell line (*Mus musculus*) | BAT cells | Shingo Kajimura **Ohno et al., 2013** | | Harvard; authenticated using methods described in **Aune et al., 2013** |
| Transfected construct (*Mus musculus*) | Ad-m-Dot1l-shRNA | Vector Biolabs | shADV-257398 | |
| Transfected construct (*Mus musculus*) | Ad-Zc3h10-flag | Vector Biolabs | ADV-276549 | |
| Transfected construct (*Mus musculus*) | Ad-m-Dot1l | Vector Biolabs | ADV-257398 | |
| Antibody | Anti-Dot1l (rabbit monoclonal) | Cell signaling | Cat# 77087 RRID:AB_2799889 | 1:1000 |
| Antibody | Anti-GAPDH (rabbit monoclonal) | Cell signaling | Cat# 5174S RRID:AB_10622025 | 1:1000 |
| Antibody | Anti-UCP1 (rabbit polyclonal) | Abcam | Cat# 10983 RRID:AB_2241462 | 1:1000 |
| Antibody | Anti-H3K79me2 (rabbit polyclonal) | Abcam | Cat# ab3594 RRID:AB_303937 | ChIP: 5 µg (5 µl) |
| Antibody | Anti-H3K79me3 (rabbit monoclonal) | Abcam | Cat# ab2621 RRID:AB_303215 | ChIP: 5 µg (5 µl) |
| Antibody | Anti-Zc3h10 (rabbit polyclonal) | Thermo Fishcer | Cat# PA5-31814 RRID:AB_2549287 | 1:1000 |
| Recombinant DNA reagent | HA-*Dot1l* expression vector | Genecopoeia | EX-Mm14067-M07 | |
| Sequence-based reagent | Dot1l_F | This paper | RT-qPCR primer | CGA CTA ATG CTG CAC CTC CT |
| Sequence-based reagent | Dot1l_R | This paper | RT-qPCR primer | AGG AGT AGT GGT GTG GCT CA |
| Sequence-based reagent | Ucp1_F | This paper | RT-qPCR primer | ACT GCC ACA CCT CCA GTC ATT |
| Sequence-based reagent | Ucp1_R | This paper | RT-qPCR primer | CTT TGC CTC ACT CAG GAT TGG |
| Sequence-based reagent | Tfam_F | This paper | RT-qPCR primer | GTC CAT AGG CAC CGT ATT GC |
| Sequence-based reagent | Tfam_R | This paper | RT-qPCR primer | CCC ATG CTG GAA AAA CAC TT |
| Sequence-based reagent | Nrf1-F | This paper | RT-qPCR primer | GAC AAG ATC ATC AAC CTG CCT GTA G |
| Sequence-based reagent | Nrf1-R | This paper | RT-qPCR primer | GCT CAC TTC CTC CGG TCC TTT G |
| Sequence-based reagent | Zc3h10-F | This paper | RT-qPCR primer | CGA CTA ATG CTG CAC CTC CT |
| Sequence-based reagent | Zc3h10-R | This paper | RT-qPCR primer | AGG AGT AGT GGT GTG GCT CA |
| Commercial assay or kit | NE-PER Nuclear and Cytoplasmic Extraction Kit | Thermo Fisher | 78833 | |
| Commercial assay or kit | SimpleChIP Plus Sonication Chromatin IP Kit | Cell signaling | 56383S | |

*Continued on next page*

*Continued*

| Reagent type (species) or resource | Designation | Source or reference | Identifiers | Additional information |
|---|---|---|---|---|
| Commercial assay or kit | Dual-Luciferase Kit | Promega | E1980 | |
| Commercial assay or kit | Nextera DNA library Preparation Kit | Illumina | 15028212 | |
| Chemical compound, drug | EPZ5676 | MedChemExpress | HY-15593 | |
| Software, algorithm | Partek Flow Genomics Suite | Partek | RRID:SCR_011860 | |
| Other | Rectal probe | Physitemp | BAT-12 | For mice |

## Animals

*Dot1l* floxed mice (*Dot1l*tm1a(KOMP)Wtsi) were generated by the trans-NIH Knock-Out Mouse Project (KOMP) and the KOMP Repository (www.komp.org). These *Dot1l* floxed mice were first mated with FLPe mice from Jackson Lab. FLP-mediated recombination was confirmed by PCR and resultant progeny were mated with *Ucp1*-Cre mice (B6.FVB-Tg(Ucp1-cre)1Evdr/J) from Jackson Lab. Unless otherwise stated, male mice between 10 and 14 weeks of age were used in experiments. Mice were fed a chow diet or a high-fat diet (HFD) (45% fat-derived calories- Dyets) ad libitum. EPZ-5676 was purchased from MedChemExpress (Cat. No.: HY-15593) and injected 20 mg/kg/day for 8 days into BAT. All protocols for mice studies were approved from the University of California at Berkeley Animal Care and Use Committee.

## GST-pulldown

*GST-Dot1l* was purchased from epicypher. *GST* and *GST-Dot1l* plasmids were transformed into BL21 (DE3)*E. coli*(NEB) and production was induced by 0.1M IPTG treatment. Resultant proteins were purified using Glutathione Sepharose 4B (GE) according to the manufacturer's recommended protocol. [$^{35}$S]-labeled Zc3h10 protein was produced by using TNT coupled transcription/translation kit (Promega). Of GST fusion proteins, 20 µg were incubated for 2 hr at 4°C with in vitro translated Zc3h10 and glutathione sepharose beads. The beads were washed three times with binding buffer, and bound proteins were eluted by boiling in Laemmli sample buffer, separated by SDS-PAGE and analyzed by autoradiography.

## Tandem affinity purification (TAP) and mass spectrometry analysis

10 ug of *Zc3h10*-TAP or TAP vector plasmid were transfected in HEK293 cells using lipofectamine 2000 (Invitrogen). Cells were lysed and immunoprecipated using buffers from the InterPlay Mammalian TAP system manufacturer recommended with the noted exception: after binding of the cell lysates to the Streptavidin resin, the resin was re-incubated with 200 µg of BAT nuclear extracts diluted to 1 mL in SBB overnight. TAP eluates were boiled in SDS, ran on an SDS-PAGE gel, and stained with Coomassie Brilliant Blue. Bands that were identified to be specific to the Zc3h10-TAP lane were excised from both the Zc3h10-TAP and TAP vector lanes and proteins were identified by mass spectrometry performed by the Vincent J. Coates Proteomics/Mass Spectrometry Laboratory (P/MSL) at UC Berkeley.

## Cold-induced thermogenesis

Core body temperature was determined using a Physitemp BAT-12 probe at 4°C.

## Indirect calorimetry

Oxygen consumption was measured using the Comprehensive Laboratory Animal Monitoring System (CLAMS). Data were normalized to lean body mass determined by EchoMRI. Mice were individually caged and maintained under a 12 hr light/12 hr dark cycle. Food consumption and locomotor activity were tracked.

## GTT and ITT

For GTTs, mice were fasted overnight, and glucose (2 mg/g) was administered intraperitoneally. For ITTs, mice were fasted 4 hr, and insulin (0.75 U/kg) was administered.

## Cell culture

HEK293 and 3T3-L1 cells were obtained from UCB Cell Culture Facility supported by The University of California Berkeley. Both cell lines were authenticated using STR analysis. Both lines were also tested for mycoplasma using a nuclear stain and a 100X lens to confirm the absence of mycoplasma in the membranes of both cell lines. The immortalized BAT cell line was from Dr. Shingo Kajimura (Harvard)(*Ohno et al., 2013*). We utilized the methods, described in *Aune et al., 2013* to verify the identity of the BAT cell line as we differentiated them and measured expression of BAT-specific genes, including *Ucp1*, *Cidea*, and *Ppargc1a*. Cells were grown in standard condition with 5% $CO_2$, at 37°C. BAT cells, 3T3-L1 cells and HEK293 cells were maintained in DMEM containing 10% FBS and 1% pen/strep prior to differentiation/transfection. Brown adipocyte differentiation was performed as described in *Yi et al., 2019*. Ad-m-*Dot1l*-shRNA, Ad-*Zc3h10*-flag adenovirus and Ad-GFP-m-*Dot1l* were purchased from Vector Biolabs. Knockdown of *Dot1l* accomplished using adenoviral transduction of MOI of 250 at day 4 of brown adipocyte differentiation. Differentiation of 3T3-L1 cells was induced by treating confluent cells with DMEM containing 10% FBS, 850 nM insulin, 0.5 mM isobutyl-methylxanthine, 1 µM dexamethasone, 1 nM T3, 125 nM indomethacin. After 48 hr of induction, cells were switched to a maintenance medium containing 10% FBS, 850 nM insulin and 1 nM T3. 3T3-L1 cells were infected on day 4 of differentiation using either GFP, *Dot1l* or *Zc3h10* adenovirus. Viral medium was replaced by maintenance medium the following day. To stimulate thermogenesis, differentiated 3T3-L1 cells were treated 6 hr with 10 µM forskolin on day 6. Inhibition of Dot1l was accomplished by addition of either 5 nM EPZ5676 (MedChemExpress, Cat. No.: HY-15593) or DMSO.

Inducible Lentiviral Nuclease hEF1α-Blast-Cas9 (GE Healthcare) was packaged into lentivirus by using MISSION Lentiviral Packaging Mix (Sigma). BAT-inducible Cas9 cell line was then generated by transduced BAT cells with inducible Cas9 lentivirus and then selected with blasticidin (10 µg/ml). Two stable *Dot1l* sgRNA expressing cell lines were generated by transducing in inducible Cas9 cells with lentivirus containing two sgRNA with target sequences of GTCTCGTGCAGCATAACCAG, or ACGCCGTGTTGTATGCATCT (Abmgood) and were then selected by neomycin (400 µg/ml). These *Dot1l* KO pools were subjected to brown adipocyte differentiation. After 48 hr of induction, cells were treated with doxycycline (1 µg/ml).

## Oxygen consumption measurement

BAT cells were differentiated in 12-well plates, trysinized, and reseeded in XF24 plates at 50 K cells per well at day 4 of differentiation and assayed on day 5 of differentiation. On the day of experiments, the cells were washed three times and maintained in XF-DMEM (Sigma-Aldrich) supplemented with 1 mM sodium pyruvate and 17.5 mM glucose. Oxygen consumption was blocked by 1 µM oligomycin. Maximal respiratory capacity was assayed by the addition of 1 µM FCCP. Tissues were incubated for 1 hr at 37°C without $CO_2$ prior to analysis on the Seahorse XF24 Analyzer. Uncoupled respiration was calculated as OCR under oligomycin treatment minus OCR under antimycin A/rotenone treatment.

## RT-qPCR analysis and western blotting

Reverse transcription was performed with 500 ng of total RNA using SuperScript III (Invitrogen). RT-qPCR was performed in triplicate with an ABI PRISM 7500 sequence detection system (Applied Biosystems) to quantify the relative mRNA levels for various genes. Statistical analysis of the qPCR was obtained using the ΔΔCt (2-ddCT) method with Eef1a1 or 18 s as the control. RT-qPCR primer sets are listed in Key resources table. Library generation for RNA sequencing was carried out at the Functional Genomics Laboratory at UC Berkeley.

For western blot analysis, total cell lysates were prepared using RIPA buffer and nuclear extracts were prepared using the NE-PER Nuclear and Cytoplasmic Extraction kit (Thermo). Proteins were separated by SDS-PAGE, transferred to nitrocellulose membrane and probed with the indicated antibodies.

## Luciferase-reporter assay

HEK293 cells were transfected with 300 ng *Zc3h10* and/or *Dot1l* expression plasmid, together with 100 ng of indicated luciferase reporter construct and 0.5 ng pRL-CMV in 24-well plates. Cells were lysed 48 hr post-transfection and assayed for luciferase activity using the Dual-Luciferase Kit (Promega) according to the manufacturer's recommended protocol.

## Plasmid constructs

The HA-*Dot1l* expression vector was purchased from Genecopoeia. The *Dot1l* sequence was subcloned by PCR amplifying and inserting into CTAP vector from Agilent, as well as pGEX-4T-3 from GE.

## Coimmunoprecipitation

For Co-IP experiments using tagged constructs, HEK293 cells were transfected using Lipofectamine 2000 to express FLAG-tagged *Zc3h10* and HA-tagged *Dot1l*. Cells were lysed in IP buffer containing 20 mM Tris, pH 7.4, 150 mM NaCl, 1 mM EDTA, 10% glycerol, 1% NP-40 supplemented with protease inhibitors. Total cell lysates were incubated 2 hr at 4°C with either anti-FLAG M2 for *Zc3h10* or anti-HA for Dot1l. For Co-IP in brown adipose tissue of wild-type mice, nuclear extraction was carried out using the NE-PER Nuclear and Cytoplasmic Extraction kit (Thermo). Equal amounts of nuclear extracts were incubated with specific antibodies and protein A/G agarose beads overnight at 4°C. Agarose beads were washed three times and bound proteins were eluted by boiling in Laemmli sample buffer and analyzed by immunoblotting using the indicated antibodies.

## ChIP-qPCR

ChIP was performed using ChIP kit (Cell signaling, 56383 s). Briefly, BAT cells overexpressing Dot1l or Zc3h10 were fixed with disuccinimidyl glutarate (DSG) at 2 mM concentration in PBS for 45 min at room temperature before cross-linking with 1% methanol-free formaldehyde in PBS for 10 min. For ChIP experiments using BAT, tissues from 3 BATs were combined and minced on ice. The reaction was stopped by incubating with 125 mM glycine for 10 min. Cells or tissues were rinsed with ice-cold phosphate-buffered saline (PBS) for three times, and lysed in IP lysis buffer containing 500 mM HEPES-KOH, pH 8, 1 mM EDTA, 0.5 mM EGTA, 140 mM NaCl, 0.5% NP-40, 0.25% Triton X-100, 10% glycerol, and protease inhibitors for 10 min at 4°C. Nuclei were collected by centrifugation at $600 \times$ g for 5 min at 4°C. Nuclei were released by douncing on ice and collected by centrifugation. Nuclei were then lysed in nuclei lysis buffer containing 50 mM Tris, pH 8.0, 1% SDS 10 mM EDTA supplemented with protease inhibitors, and sonicated three times by 20 s burst, each followed by 1 min cooling on ice. Chromatin samples were diluted 1:10 with the dilution buffer containing 16.7 mM Tris pH 8.1, 0.01% SDS 1.1% Triton X-100 1.2 mM EDTA, 1.67 mM NaCl, and proteinase inhibitor cocktail. Soluble chromatin was quantified by absorbance at 260 nm, and equivalent amounts of input DNA were immunoprecipitated using 10 µg of indicated antibodies or normal mouse IgG (Santa Cruz) and protein A/G magnetic beads (Thermo-Fisher). After the beads were washed and cross-linking reversed, DNA fragments were purified using the Simple ChIP kit (cell signaling). Samples were analyzed by qPCR for enrichment in target areas. The promoter occupancy for *Ucp1* and other indicated genes was confirmed by RT-qPCR. The fold enrichment values were normalized to input.

## RNA-seq

WT and Dot1l-BKO mice were cold exposed for 4 hr. Total RNA from BAT was prepared using RNEasy kit (Qiagen). Strand-specific libraries were generated from 500 ng total RNA using the TruSeq Stranded Total RNA Library Prep Kit (Illumina). cDNA libraries were pair-end sequenced on an Illumina HiSeq 4000. Using Partek software, reads were aligned to the mouse genome (NCBI38/mm10). A gene was included in the analysis if it met all the following criteria: the maximum RPKM reached four unit at any time point, the gene length was >100 bp. Selected genes were induced at least +/- 1.8 fold, and the expression was significantly different from the basal (p<0.05).

## ATAC-seq

For ATAC-seq, we used two mice per condition and processed nuclei extracted from BAT separately, thus each sample analyzed was a separate biological replicate. ATAC-seq was performed using Nextera DNA library Preparation kit (Illumina, 15028212) according to *Chen et al., 2018*. In all, $5 \times 10^4$ nuclei from tissues were collected in lysis buffer containing 10 mM Tris pH 7.4, 10 mM NaCl, 3 mM MgCl2, and 1% NP-40, and spun at $500 \times g$ at 4°C for 10 min. The pellets were resuspended in the transposase reaction mixture containing 25 µL $2 \times$ Tagmentation buffer, 2.5 µL transposase, and 22.5 µL nuclease-free water, and incubated at 37°C for 30 min. The samples were purified using MinElute PCR Purification kit (Qiagen, 28006) and amplification was performed in $1 \times$ next PCR master mix (NEB, M0541S) and 1.25 µM of custom Nextera PCR primers 1 and 2 with the following PCR conditions: 72°C for 5 min; 98°C for 30 s; and thermocycling at 98°C for 10 s, 63°C for 30 s, and 72°C for 1 min. Samples were amplified for five cycles and 5 µL of the PCR reaction was used to determine the required cycles of amplification by qPCR. The remaining 45 µL reaction was amplified with the determined cycles and purified with MinElute PCR Purification kit (Qiagen, 28006) yielding a final library concentration of ~30 nM in 20 µL. Libraries were subjected to pair-end 50 bp sequencing on HiSeq4000 with 4–6 indexed libraries per lane. Partek Flow Genomics Suite was used to analyze sequencing data. Reads were aligned to mm10 genome assembly using BWA-backtrack. MACS2 in ATAC mode was then used to identify peaks from the aligned reads with a q-value cutoff of 0.05 and fold enrichment cutoff of 2.0. Quantify regions tool was used to quantify the peaks identified by MACS2 from each sample to generate a union set of regions. Regions were then annotated to RefSeq mRNA database and analysis of variance (ANOVA) was performed to determine significant peak differences between groups. Gene set enrichment and pathway analysis was used to highlight biological processes with the highest q-values among those identified.

## Statistical analysis

Statistical comparisons were made using a two-tailed unpaired t-test using GraphPad Prism eight software (GraphPad Software Inc, La Jolla, CA, USA). For genome-wide analyses, we employed Partek Genomics Suite (Partek Inc, St. Louis, Missouri, USA) using ANOVA for ATAC-seq comparisons, and DESeq2 for RNA-seq differential expression comparisons, and subsequently used Gene set enrichment tool for gene ontology to identify the significantly affected pathways. Data are expressed as means ± standard errors of the means (SEM). The statistical differences in mean values were assessed by Student's *t* test. All experiments were performed at least twice and representative data are shown.

## Acknowledgements

The work was supported by NIH grant DK120075 to HSS.

## Additional information

### Funding

| Funder | Grant reference number | Author |
|--------|------------------------|--------|
| NIH Office of the Director | DK120075 | Hei Sook Sul |

The funders had no role in study design, data collection and interpretation, or the decision to submit the work for publication.

### Author contributions

Danielle Yi, Conceptualization, Resources, Data curation, Software, Formal analysis, Validation, Investigation, Methodology, Writing - original draft, Writing - review and editing; Hai P Nguyen, Software, Validation, Investigation, Methodology, Writing - review and editing; Jennie Dinh, Validation, Investigation, Methodology; Jose A Viscarra, Conceptualization, Software, Formal analysis, Methodology; Ying Xie, Data curation, Validation, Methodology; Frances Lin, Investigation; Madeleine Zhu, Methodology; Jon M Dempersmier, Conceptualization, Resources, Data curation, Investigation,

Methodology; Yuhui Wang, Formal analysis, Investigation, Methodology; Hei Sook Sul, Conceptualization, Resources, Supervision, Funding acquisition, Investigation, Project administration, Writing - review and editing

## Author ORCIDs
Danielle Yi (iD) https://orcid.org/0000-0003-3310-5151
Hei Sook Sul (iD) https://orcid.org/0000-0002-3372-5097

## Decision letter and Author response
Decision letter https://doi.org/10.7554/eLife.59990.sa1
Author response https://doi.org/10.7554/eLife.59990.sa2

# Additional files

## Supplementary files
• Transparent reporting form

## Data availability
We have deposited ATAC-seq and RNA-seq files in the NCBI database (GSE159645).

The following dataset was generated:

| Author(s) | Year | Dataset title | Dataset URL | Database and Identifier |
|---|---|---|---|---|
| Yi D, Sul HS | 2020 | Genome-wide maps of chromatin state(ATAC-seq) and gene expression(RNA-seq) of cold exposed Dot1l-ablated mouse brown adipose tissue | https://www.ncbi.nlm.nih.gov/geo/query/acc.cgi?acc=GSE159645 | NCBI Gene Expression Omnibus, GSE159645 |

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
