## [Decision Letter]

**Acceptance summary:**

This paper provides new insight into molecular mechanisms underlying the transcription of thermogenic genes in brown adipocytes by revealing an important function for Dot1l H3K79 methyltransferase activity.

**Decision letter after peer review:**

Thank you for submitting your article "Dot1L interacts with Zc3h10 to activate UCP1 and other thermogenic genes" for consideration by *eLife*. Your article has been reviewed by three peer reviewers, one of whom is a member of our Board of Reviewing Editors, and the evaluation has been overseen by Philip Cole as the Senior Editor. The reviewers have opted to remain anonymous.

The reviewers have discussed the reviews with one another and the Reviewing Editor has drafted this decision to help you prepare a revised submission.

Summary:

In this article, the authors sought to identify interacting partner(s) of Zc3h10, a transcription factor that activates expression of UCP-1 and other brown-adipocyte-specific genes. They identified the H3K79 methyltransferase Dot1L as an epigenetic modifier that interacts with Zc3h10 and facilitates its action on UCP-1 and other brown-adipocyte-specific genes. The strength of the manuscript is that the hypotheses were examined by a wide range of approaches, including protein-protein interaction assays, cell culture studies, and animal models. The present data provide solid evidence for an important role of Dot1L in the regulation of brown-adipocyte-specific genes. At the same time, all of the reviewers believed that the mechanisms involved in the epigenetic regulation of gene expression by Dot1L need to be investigated in more depth. Because the roles and the regulation of H3K79 methylation are still not fully understood, it would advance the field if the authors could provide additional data along these lines.

Essential revisions:

1) The protein expression of Dot1L appears disproportionally specific to brown adipose tissue compared with the mRNA expression. Is there a protein-gene difference? Is the identity of the band correct? Can the author see the loss of the band in the BKO tissues?

2) Most of the experiments appear to have been performed under basal conditions (e.g. unstimulated cells, room temperature mice). Dot1L expression increases with cold exposure and in their previous study that Zc3h10 is recruited to thermogenic gene promoters by p38 MAPK phosphorylation in response to adrenergic activation. The present study would benefit from additional experiments demonstrating a dynamic role for Dot1L in thermogenic gene transcription during cold exposure. Does cold exposure/adrenergic activation modulate Dot1L-Zc3h10 interaction, Dot1L recruitment to thermogenic gene promoters, H3K79 methylation, etc. in a p38 MAPK-dependent manner?

3) The authors conclude that "these results show that Dot1L methyltransferase activity is required for thermogenic gene expression and other Zc3h10 target genes for the BAT gene program" and that "Dot1L enzymatic activity is critical for activating the thermogenic gene program in vivo". These statements are not yet strongly supported. The involvement of H3K79 methylation was implied through the use of a chemical inhibitor EPZ5676 (Figure 2C and Figure 3) and the reduction of H3K79me3 levels in whole-cell lysates (Figure 3B). Can the authors confirm the change of H3K79 methylation at brown-adipocyte-specific genes by conducting ChIP-qPCR in the experiments where they modulate either expression or enzymatic activity of Dot1L? In addition, the authors could compare the effect of the overexpression of wild-type Dot1L (Figure 2D) with a mutant Dot1L in which critical residue(s) are substituted in the enzyme catalytic core.

4) In Figure 4D, ChIP-qPCR shows the reduction of H3K79me2 and H3K79me3 enrichment on thermogenic gene promoters in Dot1L knockout BAT. It would be informative to examine changes in Dot1L and Zc3h10 binding on the same regions in BAT.

5) Although not absolutely required, it would be ideal to look at global epigenetic changes following induction or inhibition of Zc3h10 and/or Dot1L. Of particular interest is whether Zc3h10 plays any role in tethering Dot1L and modulating H3K79 methylation at brown-adipocyte-specific genes and whether the epigenetic changes induced by are specific to brown adipocyte function or not. The authors should consider performing ChIP-seq for H3K79me2/3 and Zc3h10. The use of tagged protein is an alternative if the antibody is not available for the latter. H3K79me2/3 reportedly marks a subset of enhancers (Godfrey et al., 2019). It is also of interest whether such H3K79me2/3-marked enhancers are enriched in the vicinity of the brown-adipocyte-specific genes.

6) The raw ATAC-seq data show very high background (Figure 5A). How many peak calls were obtained? Improvement is necessary to discuss the qualitative differences. Please consider the use of culture cells if the technical hurdle is caused by the use of tissue samples. Further, the pattern of the aggregate plots (Figure 5A left upper) does not obviously match that of the heatmaps (Figure 5A, left lower). Please describe exactly what is shown in the figure. Are the scales the same for each panel or the heatmaps zoom in the area indicated by the red dotted boxes in the aggregate plot? What do the red dotted boxes mean? What does color scale bar in the middle heatmap mean?

7) Also related the ATAC data, Figure 5C suggests that loss of Dot1L alters chromatin accessibility at far more genes than is represented in the "Shared processes" pathway analysis. It would be helpful to see a more comprehensive analysis of the ATAC-seq data (is thermogenesis one of the top pathways, what other pathways are affected, is the Zc3h10 binding motif overrepresented at sites of increased chromatin accessibility? etc.) and discussion about why the ATAC-seq changes might be more general/less specific than the RNA-seq changes.

---

## [Author Response]

Essential revisions:1) The protein expression of Dot1L appears disproportionally specific to brown adipose tissue compared with the mRNA expression. Is there a protein-gene difference? Is the identity of the band correct? Can the author see the loss of the band in the BKO tissues?

We agree with the reviewers' comment. We repeated the western blotting (Figure 1D). There was no significant discrepancy between protein and mRNA levels of Dot1L. We also quantified the intensity of the bands of western blot (Figure 1D), which showed protein levels of Dot1L relative to GAPDH protein levels, correlating with the mRNA expression. In order to verify that the Dot1L protein band in Figure 1D is specific, we performed western blot using lysates of BAT from WT and Dot1L-BKO BAT (conditional Dot1L KO in BAT) (Figure 4A, right). Indeed, Dot1L protein level was detected highly in WT, while it was greatly lower in Dot1L-BKO confirming that the detected band indeed represented Dot1L.

2) Most of the experiments appear to have been performed under basal conditions (e.g. unstimulated cells, room temperature mice). Dot1L expression increases with cold exposure and in their previous study that Zc3h10 is recruited to thermogenic gene promoters by p38 MAPK phosphorylation in response to adrenergic activation. The present study would benefit from additional experiments demonstrating a dynamic role for Dot1L in thermogenic gene transcription during cold exposure. Does cold exposure/adrenergic activation modulate Dot1L-Zc3h10 interaction, Dot1L recruitment to thermogenic gene promoters, H3K79 methylation, etc. in a p38 MAPK-dependent manner?

All in vitro experiments in Figure 2 and in vivo experiments in Figure 3 and 4 were measured upon 6 hr forskolin stimulation or cold exposure. We now made this point clear in the manuscript and figure legends. Furthermore, we examined the Dot1L-Zc3h10 interaction in cold-exposed mouse BAT by CoIP (Figure 1—figure supplement 1C). BAT lysates from mice housed at room temperature or 4°C were immunoprecipitated with either IgG or Dot1L and immunoblotted with Zc3h10. We previously showed that the binding of Zc3h10 to the UCP1 promoter is increased in a cold exposure/p38 MAPK-dependent manner (Yi et al., 2019). However, as we did not detect any difference between conditions, Dot1L-Zc3h10 interaction is not dependent on cold exposure.

3) The authors conclude that "these results show that Dot1L methyltransferase activity is required for thermogenic gene expression and other Zc3h10 target genes for the BAT gene program" and that "Dot1L enzymatic activity is critical for activating the thermogenic gene program in vivo". These statements are not yet strongly supported. The involvement of H3K79 methylation was implied through the use of a chemical inhibitor EPZ5676 (Figure 2C and Figure 3) and the reduction of H3K79me3 levels in whole-cell lysates (Figure 3B). Can the authors confirm the change of H3K79 methylation at brown-adipocyte-specific genes by conducting ChIP-qPCR in the experiments where they modulate either expression or enzymatic activity of Dot1L? In addition, the authors could compare the effect of the overexpression of wild-type Dot1L (Figure 2D) with a mutant Dot1L in which critical residue(s) are substituted in the enzyme catalytic core.

We have already shown that decreased Dot1L expression in Dot1L-BKO mice significantly reduces H3K79me2/3 at thermogenic genes by ChIP-qPCR using chromatin from WT and Dot1L-BKO mice in Figure 4D. In order to confirm that the decrease of thermogenic gene expression is due to the enzymatic activity of Dot1L, we overexpressed Dot1L in differentiated BAT cells and treated with EPZ5676, Dot1L inhibitor (10uM), then performed ChIP-qPCR for H3K79me3. EPZ5676 is a highly selective Dot1L inhibitor as it occupies the S-adenosyl methionine (SAM) binding pocket and inhibits the enzymatic activity of Dot1L (Diagle et al., 2013). EPZ5676 treated cells had significantly decreased H3K79me3 at -4.6kb UCP1 region as well as Tfam and Nrf1 promoter regions, while having no difference at the GAPDH promoter region as compared to the control cells (Figure 2D).

Moreover, as the reviewer proposed, we generated M55L-Dot1L mutated of a critical residue in the catalytic core. M55 is located at the N-terminal domain reported to be critical for histone substrate binding site and affects demethylase catalytic activity (Zhu et al., 2018). We overexpressed WT-Dot1L as well as the Dot1L mutant in BAT cells and performed ChIP-qPCR (Figure 2—figure supplement 1C). Dot1L-M55L mutant significantly reduced H3K79me3 at the promoter regions of UCP1, Tfam and Nrf1, compared to WT-Dot1L, while not affecting GAPDH (Figure 2—figure supplement 1C). Taken together, based on ChIP-qPCR in the experiments in which expression or enzymatic activity of Dot1L is modulated using both the inhibitor and the mutant, the enzymatic activity of Dot1L is critical for increasing H3K79 methylation at thermogenic genes, thereby increasing thermogenic gene expression.

4) In Figure 4D, ChIP-qPCR shows the reduction of H3K79me2 and H3K79me3 enrichment on thermogenic gene promoters in Dot1L knockout BAT. It would be informative to examine changes in Dot1L and Zc3h10 binding on the same regions in BAT.

We thank the reviewers for this comment. In order to show that Dot1L and Zc3h10 bind on the same regions of thermogenic genes, we performed reChIP-qPCR using Zc3h10 and Dot1L antibodies sequentially (Figure 1H, right). We observed significant enrichment of Dot1L for -4.6 kb region of the UCP1 promoter when we first used Zc3h10 antibody followed by Dot1L antibody, and this enrichment was similar to that of Zc3h10 alone. There was no enrichment observed in the control, GAPDH promoter.

5) Although not absolutely required, it would be ideal to look at global epigenetic changes following induction or inhibition of Zc3h10 and/or Dot1L. Of particular interest is whether Zc3h10 plays any role in tethering Dot1L and modulating H3K79 methylation at brown-adipocyte-specific genes and whether the epigenetic changes induced by are specific to brown adipocyte function or not. The authors should consider performing ChIP-seq for H3K79me2/3 and Zc3h10. The use of tagged protein is an alternative if the antibody is not available for the latter. H3K79me2/3 reportedly marks a subset of enhancers (Godfrey et al., 2019). It is also of interest whether such H3K79me2/3-marked enhancers are enriched in the vicinity of the brown-adipocyte-specific genes.

We have done ChIP-seq for Zc3h10 in our previous paper and have shown that Zc3h10 binds to -4.6 kb UCP1 promoter region as well as Tfam and Nrf1 (Yi et al., 2019). Although it would be interesting to examine the global epigenetic changes H3K79 methylation by Dot1L, we showed by reChIP-qPCR that Dot1L and Zc3h10 occupy the same regions of thermogenic genes, such as -4.6 kb UCP1 and +50 bp Tfam (Figure 1H). In addition, we showed a decrease of H3K79me2/3 in Dot1L-BKO mice in the same regions of theses thermogenic gene promoters, indicating that Dot1L is recruited by Zc3h10 to the vicinity of the brown-adipocyte-specific genes.

6) The raw ATAC-seq data show very high background (Figure 5A). How many peak calls were obtained? Improvement is necessary to discuss the qualitative differences. Please consider the use of culture cells if the technical hurdle is caused by the use of tissue samples. Further, the pattern of the aggregate plots (Figure 5A left upper) does not obviously match that of the heatmaps (Figure 5A, left lower). Please describe exactly what is shown in the figure. Are the scales the same for each panel or the heatmaps zoom in the area indicated by the red dotted boxes in the aggregate plot? What do the red dotted boxes mean? What does color scale bar in the middle heatmap mean?

We agree with the reviewers that our ATAC-seq data had high background. During this revision, we re-performed ATAC-seq with WT and Dot1L-BKO BAT. At this time, we cold-exposed mice for 4 hrs, longer than the previous time to demonstrate an obvious difference in the chromatin structure between samples. Our analyses found approximately 20,000 peaks per condition (WT and Dot1L-BKO). For the WT sample, we detect a high number of peaks in the aggregate plot (upper) and high ATAC-seq read distributions concentrated at the TSS in the heatmap for the WT (lower), while Dot1L-BKO had significantly reduced peaks near the transcription start sites (TSS), signifying reduced open chromatin regions. The pattern of the aggregate plots now match that of the heatmaps. Scales within each panel are the same, for the ATAC-seq heatmaps the color scale indicates the number of individual genes associated with the peaks identified. With our primary interest in assessing Dot1L’s role in the regulation of thermogenesis, we compared ATAC-seq peaks for thermogenic genes, such as UCP1, Cox8b and Cox7a1 in response to Dot1L ablation (Figure 5A right). The dotted boxes were used to point out the difference in peak intensity in WT and Dot1L-BKO, and nothing is modified or zoomed in.

7) Also related the ATAC data, Figure 5C suggests that loss of Dot1L alters chromatin accessibility at far more genes than is represented in the "Shared processes" pathway analysis. It would be helpful to see a more comprehensive analysis of the ATAC-seq data (is thermogenesis one of the top pathways, what other pathways are affected, is the Zc3h10 binding motif overrepresented at sites of increased chromatin accessibility? etc.) and discussion about why the ATAC-seq changes might be more general/less specific than the RNA-seq changes.

To gain comprehensive understanding of the genome wide chromatin structure changes, we now examined the pathways that are affected by Dot1L ablation in BAT using Gene Ontology, instead of picking out individual genes. Some of the most affected pathways are cell cycle, chromatin organization, mitochondrion organization, thermogenesis etc. (Figure 5A middle). Thermogenesis is not one of the top pathways, however, it is one of the significant. Based on the motif analysis, a number of motifs are CC doublets or GC rich similar to the Zc3h10 binding motif, TYCCNG, that we identified from SELEX data and ChIP-seq data in the WT samples (Yi et al., 2019). While this motif is not yet present in the Partek flow database, there are several similar motifs, such as that for E2F8 (TTTCCCG) and TEAD3 (TWCCD), that are overrepresented in open chromatin regions in the WT and not the Dot1L-KO. In fact, we detected a significant difference in CC doublets or GC rich, which are similar to the Zc3h10 binding motif, between the WT and KO samples. ATAC-seq demonstrates open chromatin regions which can help predict increased transcription because it creates an environment permissive to transcription factor recruitment while RNA-seq actually shows the transcripts being produced.